# Lack of caspase 8 directs neuronal progenitor-like reprogramming and small cell lung cancer progression

Ariadne Androulidaki[1,2], Fanyu Liu[1,2], Christina M. Bebber [1,2], Ilmars Kisis [1], Vignesh Sakthivelu [1], Pascal Hunold [3], Lioba Koerner [2,4], Alina Dahlhaus [1,2], Fatma Isil Yapici [1,2], Christina Grimm [3,5], Alicja Pacholewska [3,5], Sofya Tishina [1,2], Franka Doskotz [1,2], Lucia A. Torres Fernández [1], Jenny Stroh[1,2], Ali T. Abdallah[2,6,7], Julia Beck[1,2], Lejla Mulalic[1,2], Anna Schmitt[2,8], Holger Grüll[9], Thorsten Persigehl[9], Alexander Quaas[10], Martin Peifer [1,3], Johannes Brägelmann [1,3,11], H. Christian Reinhardt [12], Pascal Nieper[1,8], Robert Hänsel-Hertsch [3], Roman K. Thomas [1,10], Julie George [1,13], Michal R. Schweiger [3,5], Manolis Pasparakis [2,3,4], Filippo Beleggia [1,8,11] & Silvia von Karstedt [1,2,3] ✉

Most neuroendocrine cancers lack caspase 8 protein expression. While this feature was thought to facilitate escape from extrinsic apoptosis, its cancer-regulatory function has remained unexplored. Here, we devise a mouse model of small cell lung cancer (SCLC) recapitulating the lack of expression of caspase 8 seen in humans and uncover an unexpected role for necroptosis-fueled pre-tumoral inflammation resulting in reprogramming towards a neuronal progenitor cell-like state and increased metastatic disease. Notably, transcriptional signatures of this cellular state are enriched in relapsed and metastatic human SCLC. Mechanistically, caspase 8 loss within the pre-tumoral niche promotes inflammation marked by increased recruitment of regulatory T cells (Tregs) which are responsible for the promotion of metastatic disease. Importantly, inactivation of the necroptosis executioner MLKL reverses pre-tumoral inflammation, decreases metastasis as well as neuronal-like reprogramming. Taken together, our findings suggest that pre-tumoral inflammatory cell death contributes to neuronal progenitor mimicry, immunosuppression and increased metastasis in SCLC.

Small cell lung cancer (SCLC) is amongst the most aggressive forms of lung cancer with a 5-year survival rate of only 5%[1]. Despite this dismal prognosis, SCLC is initially highly sensitive to chemotherapy in most cases, but patients inevitably relapse with very rapid progression. SCLC unlike other epithelial cancers is thought to arise from a distinct cellular population of pulmonary neuroendocrine cells (PNECs) which share features with neuronal cells[2,3]. Normal neural stem or progenitor cells were shown to lack caspase 8 expression[4] suggesting this to be a characteristic neuroendocrine (NE) feature. Indeed, lack of caspase-8 expression is frequently observed in other NE cancer entities, including neuroblastoma, medulloblastoma, and glioblastoma[5]. Human SCLC, but not non-small cell lung cancer (NSCLC) cell lines, lacked caspase 8 expression and, as a result of this, were resistant to induction of extrinsic apoptosis[6]. DNA hypermethylation within the caspase 8 gene promoter region was found in SCLC primary tumors and carcinoid tumor samples, indicating epigenetic rather than genetic loss[7].

**Fig. 1 | Lack of caspase 8 expression promotes metastatic disease. a** Caspase 8 protein expression (log[2]) of SCLC patients (n = 112) and matched adjacent normal lung samples (n = 112)[18] is plotted. **b** Caspase 8 expression transcripts per kilobase million (TPM) (log[2] TPM+1) in lung adenocarcinoma (TCGA; LUAD, n = 48) and SCLC (n = 81) samples is shown. **c** Spearman correlation of CASP8 expression (FPKM) versus the chromosomal status of CASP8 determined as integral copy number (iCN) for n = 71 patients with SCLC[20]. **d** Methylation determined for 33 patient-matched tumor and normal samples by MeDIP-seq. CASP8 promoter methylation levels (average methylation of ten different regions) are plotted. **e** Pearson correlation determined for CASP8 methylation (CASP8 methylation mean region 4–8) and CASP8 NM_033356 expression (FPKM). **f** Expression levels (log[2] TPM+1) of caspase 8 in human (n = 14) and mouse SCLC (n = 10), human (n = 77) and mouse NSCLC (n = 5) cell lines. **g** Tumor incidence of RP mice (n = 19) and RPC mice (n = 21) based on first detection of tumors using MRI imaging.

**h** Representative MRI images and photos of lungs of RP mice (n = 26) and RPC mice (n = 24) sacrificed at humane endpoint. Quantification of the number of tumor nodules per lung based on macroscopical evaluation at time of death.
**i** Representative pictures of liver metastasis at humane time point of RP mice (n = 24) and RPC mice (n = 16) and quantification of liver metastasis incidence at endpoint based on macroscopical evaluation. **j** Survival of mice at 40 weeks after adenoCre inhalation (RP n = 13 and RPC n = 17). **k** qPCR analysis on endpoint tumors from RP (n = 5) and RPC (n = 5) mice. Expression values are depicted as relative expression (DCt) to housekeeping gene. **l** Representative images of lung sections from mice with the indicated genotypes sacrificed at humane endpoint, stained with H&E or immunostained for Ki67, NCAM (CD56), CD31 or CD45 are shown. Scale bars: 100 μm (RP n = 6 and RPC n = 10). **a, b, d, f** paired t tests (two tailed). **c** Spearman correlation (two-tailed). **e** Pearson´s correlation (two tailed). **g–j** Fisher´s exact test (two-sided). Source Data are provided as a Source Data file.

Notably, loss of caspase 8 expression, but not heterozygous loss, renders cells insensitive to extrinsic apoptosis[8,9]. In addition to blunting extrinsic apoptosis, absence of caspase 8 expression can instead allow for necroptosis to occur upon death receptor ligation[10]. Necroptosis, unlike apoptosis, results in plasma membrane rupture via mixed lineage kinase domain like pseudokinase (MLKL)-mediated pore formation[11,12] and, hence, is an inflammatory type of cell death with reported anti-tumor activity[13,14]. Yet, necroptosis-driven inflammation can also have profound pro-tumor functions[15] and, in this respect, necroptosis has indeed also been reported to promote tumor development[16]. Despite these insights in other contexts, how absent caspase 8 expression shapes NE tumor development, biology and progression in SCLC has remained unexplored. Here, we show that pretumoral necroptosis induced by lack of caspase 8 in a mouse model for SCLC promotes neuronal progenitor-like reprogramming and immunosuppression that enhances metastasis and resembles aggressive human SCLC.

## Results

### Epigenetic silencing of caspase 8 is a characteristic feature of SCLC facilitating progression to metastatic disease

We have previously shown that caspase 8 mRNA expression is absent in the majority of an SCLC patient cohort, compared to normal lung[17]. To determine whether low caspase 8 protein expression is a distinctive feature of primary SCLC, we analyzed a recently published proteogenomic dataset including paired tumors and adjacent lung samples from 112 treatment-naïve SCLC patients[18]. Indeed, caspase 8 protein levels in SCLC tissue were also much lower than in matched adjacent normal lung (Fig. 1a). This difference was also evident on RNA level, albeit less pronounced (Supplementary Fig. 1a). Interestingly, using bioinformatic tumor purity estimates[19] from whole genome sequencing (WGS) data of human SCLC patient samples (n = 70)[20], we found that tumor purity negatively correlates with detected caspase 8 expression within human SCLC biopsies indicating that residual caspase 8 expression detected in bulk RNA-sequencing data from a

minority of patients likely derives from non-tumor contaminants (i.e., immune cell, normal lung cells) (Supplementary Fig. 1b). Next, we compared caspase 8 expression in bulk-RNA-sequencing data from SCLC with lung adenocarcinoma (LUAD). Indeed, tumors from patients with SCLC displayed significantly lower caspase 8 expression than LUAD patient samples (Fig. 1b). In order to determine whether low caspase 8 expression was a result of genetic loss in SCLC patient tissue, we analyzed the genomic locus of human caspase 8 (CASP8) from SCLC WGS data[20]. Yet, mutations in CASP8 are rare ($n = 2/110$[20]), chromosomal losses encompassing the locus of CASP8 occurred only in 5 cases and we found no correlation with CASP8 transcript levels. This suggested that low expression of CASP8 in most patients might derive from epigenetic rather than genetic alterations (Fig. 1c). We therefore performed methylome studies on these tumors, and indeed, when analyzing the CASP8 locus in primary SCLC samples obtained through surgical resections from 33 patients diagnosed with stage I–IV SCLC (Supplementary Data 1) together with their respective normal adjacent tissue, we found significantly elevated methylation in 10 regions across the CASP8 gene (Supplementary Fig. 1c), 5 of which (region 4–8) were in a regulatory region identified by the ENCODE project close to the promoter region of a shorter isoform of CASP8 (ENST00000323492) -one of the two major isoforms expressed within normal lung[21]- and the major isoform expressed in a variety of cancers, especially LUAD[22] (Fig. 1d). Importantly, increased methylation within the promoter region correlated with decreased expression of caspase 8 in bulk RNA-sequencing data of 33 patients (Fig. 1e), suggesting that promoter hypermethylation and ensuing lack of expression of CASP8 define a characteristic feature of primary human SCLC. In line with the above, we could show that, human SCLC cell lines representing various transcriptional subtypes[1], express less caspase 8 mRNA than human NSCLC cell lines (Fig. 1f). Interestingly however, adherent cell lines with non-neuroendocrine differentiation derived from the widely used genetically engineered mouse model for SCLC (RP-mice)[23] did not show lower caspase 8 mRNA expression than cell lines derived from Kras$^{G12D}$/Trp53-deletion driven NSCLC (KP-mice)[24] despite the same genetic background of both mouse models (C57BL/6) (Fig. 1f). Moreover, while caspase 8 protein levels were almost undetectable in human SCLC cell lines as compared to human NSCLC cell lines (Supplementary Fig. 1d), caspase 8 protein was expressed at comparable levels between murine NSCLC and SCLC cell lines (Supplementary Fig. 1e), indicating that the RP mouse model retains high expression levels of caspase 8. Therefore, we find that, while these data identify lack of caspase 8 protein expression to be a characteristic feature of human SCLC, this is currently not faithfully recapitulated in a commonly used genetically engineered mouse model for SCLC.

To understand the functional relevance of low caspase 8 expression in SCLC, we mimicked this human feature of SCLC by crossing RP-mice to animals with conditional caspase 8 knockout (C8$^{FL/FL}$)[25] to study its role in SCLC tumor onset, metastasis and survival. To this end, groups of 8–12-week-old RP-C8$^{WT/WT}$ and RP-C8$^{FL/FL}$ mice (called RP and RPC from hereon) were intratracheally inhaled with adenoviral CMV-Cre (Ad-Cre) to initiate lung tumor development through combined Rb1/Trp53 deletion with or without concomitant Casp8 deletion. Mice were regularly monitored for tumor development by magnetic resonance imaging (MRI) to determine the time of tumor onset. The first tumors appeared at the same time for both genotypes (around 18 weeks post inhalation both some RP and some RPC mice show tumors). Interestingly, however, a higher percentage of RPC mice presented with tumors at 23–27 weeks post inhalation. At week 24 post Ad-Cre inhalation only 37% of RP mice had MRI-confirmed lung tumors, more than 67% of RPC mice showed signs of tumor development in the lung (Fig. 1g). In addition to the observed increased percentage of mice with tumors at this time, RPC mice developed significantly more individual tumor nodules per lung indicative of either multiple primary tumor-initiating events or an increase in

secondary lung metastasis (Fig. 1h). Indeed, at the experimental end-point, we observed an almost doubled incidence in liver macro-metastasis in RPC mice (Fig. 1i). Possibly as a result of differences in primary tumor size versus metastatic burden determining the end-point in RP versus RPC mice, Kaplan Meier survival did not significantly differ between both groups (Supplementary Fig. 1f). At 40 weeks post inhalation however, there was a trend towards less RPC mice being alive (Fig. 1j). Quantitative real-time PCR (qPCR) bulk analysis of common SCLC molecular subtype markers (ASCL1, NEUROD1, POU2F3, YAP1; SCLC-A/N/P/Y) within RP and RPC tumors showed no overall significant difference in marker expression and a predominant expression of ASCL1 in both genotypes (Fig. 1k). Histological analysis (H&E) of endpoint tumors from RP and RPC mice confirmed typical SCLC morphology with dense sheets of small cells with scant cytoplasm, aberrant necrosis and high proliferation in both groups further indicating that the tumors that arise in RPC mice retain SCLC characteristics. In addition, tumors were positive for the NE markers ASCL1 and NCAM (Fig. 1l). RPC tumors showed abundant tumor stroma according to pathologist evaluation. Interestingly, RPC tumors also showed increased presence of rosette-like structures (Supplementary Fig. 1h), a histologic microanatomical pattern commonly seen in human NE tumors[26] and also observed during in vitro differentiation of pluripotent stem cells[27]. Both RP- and RPC- tumors presented with an immune cell exclusion phenotype wherein immune cells were restricted to the periphery or evident near blood vessels (Fig. 1l and Supplementary Fig. 1i), a feature frequently observed in human SCLC tumor tissue[28]. Additionally, both RP and RPC tumors showed aberrant CD31 and phosphorylated γ-H2AX-positive cells, indicating the presence of angiogenesis and DNA damage, respectively, without significant differences between the two genotypes (Supplementary Fig. 1g, j). Taken together, these results provide evidence that lack of caspase 8 expression is an inherent feature of the majority of human SCLC and recapitulation of this feature in a genetically engineered mouse model reveals that it serves progression to metastatic disease.

## Lack of caspase 8 expression promotes reprogramming towards a neuronal progenitor-like state

To obtain an in-depth characterization of RPC tumors, we next performed single-nucleus RNA-sequencing (snRNA-seq) from fresh-frozen RP and RPC tumors from two individual mice in each case at endpoint. After initial quality control, we obtained transcriptomes from 17,188 cells from RP and 21,014 cells from RPC tumors. To distinguish tumor from non-tumor cells in our samples, we first mapped all samples to an annotated single-cell reference dataset[29] containing healthy C57BL/6JN mouse lung and trachea cells. Cells clustering together with the reference dataset were assigned as normal lung cells. Clusters filled predominantly (threshold 95%) by query samples were assigned as tumor cells (Supplementary Fig. 2a, b). Clustering of tumor cells identified 13 distinct populations which were plotted using Uniform Manifold Approximation and Projection (UMAP) (Fig. 2a) with different proportions of RP and RPC cells (Fig. 2b and Supplementary Fig. 2c). In line with the bulk qPCR analysis (Fig. 1k), the expression of common transcriptional subtype markers of SCLC (A/N/P/Y) did not differ between RP and RPC tumors, and ASCL1 seemed to be the predominantly expressed transcription factor in both genotypes (Supplementary Fig. 2d). Importantly however, RP and RPC clusters did not completely overlap suggesting distinct features characterizing RPC tumors (Fig. 2c). To identify unique characteristics of RPC tumors we determined gene ontology (GO) pathways from genes enriched in the 6 clusters dominated by RP cells (clusters: 2, 4, 5, 8, 9, 12) as compared to genes from the 4 clusters dominated by RPC cells (clusters: 0, 1, 11, 13) using STRING analysis. While most RP clusters were strongly enriched in "Nervous system development" GO terms, RPC-dominated clusters were positively enriched in GO terms related to "(ion) transport" (Supplementary Fig. 2e). Interestingly, the GO term

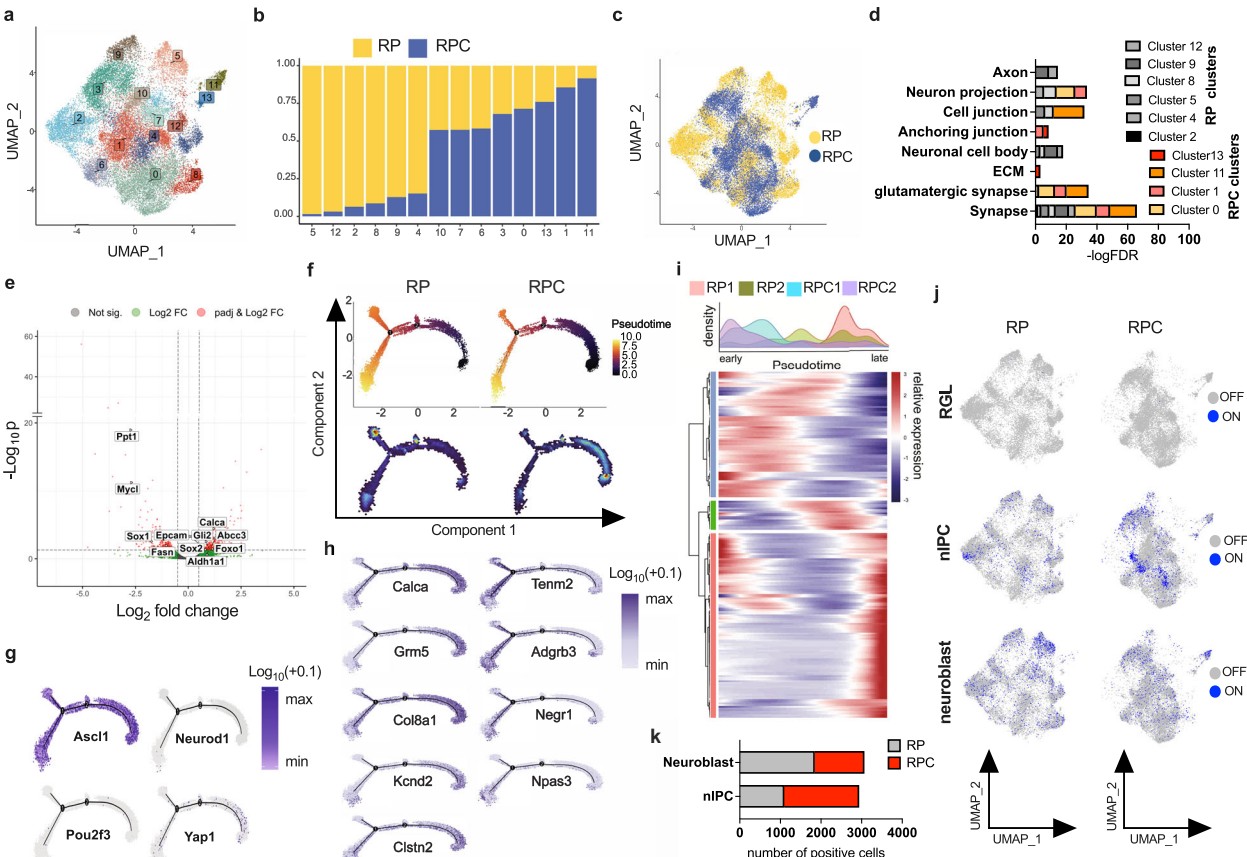

**Fig. 2 | Absence of caspase 8 promotes reprogramming towards a neuronal progenitor-like state. a** Uniform Manifold Approximation and Projection (UMAP) clustering of tumor cells (n = 28,000) from RP (n = 2 mice n = 7000 cells) and RPC (n = 2 mice, n = 7000 cells) samples. **b** Proportions of RP (yellow) and RPC (blue) cells within each cluster identified. **c** UMAP plot of RP (yellow) and RPC (blue) clusters. **d** Pathway enriched in STRING analysis on the upregulated cluster-defining genes are shown. Top enriched pathways within GO term ´Cellular Component´ plotted based on False Discovery Rate (FDR). **e** Volcano plot of differentially expressed genes (pseudo bulk) in RPC as compared to RP tumor cells.

**f** Unsupervised pseudotime trajectory analysis of RP and RPC tumor cells (upper panel), density plots (dark colours=lower density) (lower panel). **g** Expression of SCLC-A/N/P/Y subtype genes on the trajectory. **h** Genes defining early (left) or late (right) pseudotime. **i** Heat map of pseudotime-defining gene clusters and sample distribution along pseudotime in density histograms. **j** Binary AUCell score activity projected onto RP (left) and RPC (right) UMAP space. **k** Number of RP (grey) and RPC (red) cells with binary AUCell score activity=on is plotted. Source Data are provided as a Source Data file.

´Glutamatergic synapse´ was enriched in 3 out of 4 RPC-dominated clusters and only in one of the RP clusters (Fig. 2d). Gene set enrichment analysis from a ranked list between the two "extreme" clusters (5 versus 11) also showed a strong enrichment of the "Glutamatergic synapse" gene set in RPC tumors (cluster 11) (Supplementary Fig. 2f, g). Pseudobulk RNA expression in RPC tumors presented with significantly upregulated pathways such as "Cilium movement" and "Axoneme assembly" (Supplementary Fig. 3a), while significantly upregulated genes in RPC tumors, included genes promoting stem cell renewal and inhibiting neuronal differentiation, such as Foxo1, Gli2[30], Sox-2 and Aldh1a1, or the ABC transporter Abcc3 (Fig. 2e). Notably, Calca, encoding the Calcitonin- Related Peptide (CGRP), a marker of PNECS, but also marker of an SCLC-A NE variant, SCLC-A2 or NEv2[31,32], was also significantly upregulated in RPC tumors. Vice versa, genes that promote neuronal differentiation and known to maintain neuronal identity in SCLC including Sox1 and MycL, but also Dll3, an inhibitory Notch ligand, were significantly downregulated in RPC tumors (Fig. 2e). Accordingly, the significantly downregulated genes within pseudo-bulk mRNA expression showed a strong enrichment in "Nervous system development" (Supplementary Fig. 3b). Of note GO terms enriched in RPC tumors such as "Glutamatergic synapse" and "Ion Channel signaling", have been shown to be highly enriched in very early stages of neurogenesis[33]. This distinct expression pattern led us to hypothesize that RPC tumor cells might phenocopy an "earlier" step

within a differentiation trajectory mimicking neuronal progenitor-to-neuron differentiation. To test this, we performed unsupervised pseudotime trajectory analysis comparing RP and RPC tumors. Remarkably, RPC cells strongly occupied "early" pseudotime on the trajectory, while RP cells were enriched further along the trajectory and at distal branchpoints (Fig. 2f). Additionally, RP- and RPC-cell dominated clusters showed a clear difference in pseudotime progression with RPC clusters (clusters: 0, 1, 13) being enriched in early pseudotime (Supplementary Fig. 3c). Of note, ASCL1 was uniformly expressed throughout the trajectory while the expression of NEU-ROD1, POU2F3 or YAP1 was neglectable (Fig. 2g), indicating that SCLC-A/N/P/Y subtypes do not drive directionality of this trajectory. Moreover, in a recently described single cell RNA-seq dataset depicting temporal plasticity from ASCL1+ mouse SCLC to non-NE YAP1+ SCLC[34], caspase 8 expression also did not follow a clear SCLC-A/N/P/Y subtype pattern (Supplementary Fig. 3d). Instead, Calca, the metabotropic glutamate receptor Grm5, as well as genes positively associated with angiogenesis and metastasis such as Col8a1 and ion channels like Kcnd2, seemed to be genes within "early" pseudotime. Interestingly, the other end of the trajectory was defined by genes involved in neuronal differentiation and development (Tenm2, Adgrb3, Npas3 or Negr1) (Fig. 2h). In support, enrichment analysis of the 200 most differentially expressed genes that were used to construct the trajectory again revealed significant enrichment in pathways such as "Nervous

system development", "Generation of neurons", "Neuron differentiation" and "Neurogenesis" (Supplementary Fig. 3e). To resolve changes in these genes over (pseudo)time and by mouse sample and genotype, we performed unsupervised heatmap clustering. This analysis showed three clusters of genes enriched in early-to-intermediate (blue), later (green) and late/late and early (red) pseudotime. Remarkably, both RPC samples showed an elevated density in early pseudotime gene clusters while both RP samples accumulated further along pseudotime in intermediate and late clusters (Fig. 2i). These data supported the hypothesis that lack of caspase 8 expression favors SCLC tumors which retain NE (ASCL1⁺) features but are enriched in cell states which phenocopy earlier states of neuronal differentiation. Interestingly, ASCL1 directly controls the specification of neuronal progenitors as well as the later steps of neuronal differentiation[35]. During adult neurogenesis quiescent neuronal stem cells, also referred to as radial glia-like cells (RGLs), become activated, enter the cell cycle, and generate neuronal intermediate progenitor cells (nIPCs) which in turn give rise to the more lineage-committed neuroblasts that will further differentiate into immature and then mature neurons[33]. To further test the hypothesis that lack of caspase 8 expression favors a cell state mimicking earlier stages of neurogenesis, we generated AUCell[36] scores from single-cell expression data of normal RGLs, nIPCs and neuroblasts using a dataset tracing normal dentate gyrus neurogenesis in postnatal mouse development[37]. Binary AUCell score activity was projected onto UMAP space of RP and RPC tumors. Indeed, RPC tumors showed an enrichment of tumor cells with active nIPC AUCell score. Vice versa RP tumors contained more cells with active neuroblast AUCell score (Fig. 2j, k). Collectively, these data show that lack of caspase 8 expression in SCLC results in less differentiated and more stem-cell like NE tumors mimicking reprogramming towards a neuronal progenitor-like state.

### Lack of caspase 8 expression promotes epigenetic reprogramming towards a stem cell-like state and phenocopies transcriptional profiles of relapse SCLC

To further validate these findings and test whether this cellular state is retained in the absence of a tumor microenvironment, we isolated cell lines from RP and RPC tumors and performed bulk RNA-Seq analysis on them. RPC cell lines maintained absence of caspase 8 protein expression, accompanied by loss of sensitivity to undergo extrinsic apoptosis (Supplementary Fig. 4a–e). Expression of NE (ASCL1) and non-NE markers (REST1 and YAP1) was found to vary between cell lines, but no significant overall difference was observed between the two genetic groups (Supplementary Fig. 4f–i). Remarkably, among the top 50 upregulated genes in RPC-derived cell lines we found genes involved in maintaining a neuronal stem- cell state such as Six3, Pax3, Dlk1 and Nr2f1, as well as genes that are commonly expressed in stem cells like the multi-drug resistant marker Abcc3 (Supplementary Fig. 5a). Accordingly, within the top significantly downregulated genes we identified Doublecortin (Dcx), a master regulator of neurogenesis and marker of neuroblasts/neuronal progenitors, as well as other important regulators of neuronal development including Mafb (Supplementary Fig. 5a). In line with pathways regulated in our ex vivo single cell analysis from fresh-frozen tissue, GO term analysis on genes significantly upregulated in RPC-derived cell lines revealed enrichment in pathways related to "neurogenesis", but also in pathways related to metastasis like "cell migration", "angiogenesis" and "cell adhesion" (Fig. 3a). Vice versa, we observed a strong enrichment of pathways related to "Nervous system development" and "Neuron differentiation" in the genes that were significantly downregulated in RPC cell lines (Fig. 3b). qPCR analysis confirmed the differential expression of genes related to neuronal differentiation which were found in the top 50 "up" or "down" regulated genes, like Six3, Pax3, Dlk1 or Dcx, but also Calca and Sox1, which were identified within snRNA-seq of tumors and strongly relate to the neuronal identity of SCLC (Supplementary

Fig. 5b). These data suggest that expression profiles of RPC-derived cell lines very much echo transcriptional profiles of freshly isolated RPC single cells coming from an intact microenvironment.

Next, to determine the extent of epigenetic reprogramming in RPC-derived cells, we subjected them to Methyl-seq analysis. RPC tumor-derived cell lines presented with significantly altered overall methylation patterns, resulting in 1171 differentially-methylated regions (DMRs) from which 552 located to putative promoter regions (Fig. 3c). STRING analysis showed significant DNA hyper-methylation again in pathways related to "Nervous system development", "Neurogenesis" or "Neuron differentiation" (Supplementary Fig. 5c), while significant DNA hypo-methylated genes were vice versa enriched in pathways related to "Negative regulation of cell differentiation", "Negative regulation of metabolic process", but also to "Cell migration" and "Focal adhesion assembly" (Supplementary Fig. 5d). Importantly, pathways showing significant hypomethylation showed a significant overlap with pathways significantly upregulated within differentially expressed genes, and, similarly, the hypermethylated pathways with the downregulated ones, suggesting epigenetic reprogramming towards a more stem- cell- like state in RPC cells (Fig. 3d, e and Supplementary Fig. 5e, f).

To functionally assess the enhanced stem-cell-like state in RPC cells, we next tested their capacity to initiate spheroids. Indeed, spheroid formation capacity was significantly enhanced in RPC as compared to RP-derived cell lines (Fig. 3f). Interestingly, expression of common stem-cell-like genes were upregulated over time in RPC spheroids sampled at 20 days as compared to spheroids sampled at 40 days suggesting further adaptation processes (Supplementary Fig. 6a). Moreover, RPC cells demonstrated increased pluripotency when forced to differentiate into unrelated cell types such as osteocytes (Fig. 3g) or adipocytes (Supplementary Fig. 6b). Given that stem-like cancer cells are known to be more resistant to chemotherapy, we tested this aspect also in RP as compared to RPC cells. Indeed, RPC-derived cell lines were more resistant to cisplatin/etoposide (cis/eto) treatment as compared to RP-derived cells (Fig. 3h, and Supplementary Fig. 6c). To next determine whether transcriptional signatures defining mouse RPC tumors can be found within human ASCL1⁺ patient samples, we made use of a recently obtained human SCLC patient dataset containing annotated samples from various stages pre- and post-treatment[38]. Interestingly, expression of genes significantly upregulated in RPC tumors was overall higher in clusters enriched in relapsed and metastatic patients (Fig. 3i). While a few treatment-naïve patients already had metastasis at diagnosis, metastasis occurrence was vastly enriched within the relapse cohort that correlated with RPC transcriptional signatures. Collectively, these results show that RPC-derived cell lines retain a stem-cell-like state which resembles highly aggressive and metastatic human SCLC upon relapse.

### Exacerbation of pre-tumoral immunosuppressive inflammation is responsible for enhanced metastatic disease caused by absence of caspase 8

To determine whether deletion of caspase 8 expression in SCLC cells ex vivo would be sufficient to recapitulate stem-cell like marker expression observed in RPC-derived cells we suppressed caspase 8 expression in RP-derived cells using siRNA. However, siRNA-mediated silencing of caspase 8 was insufficient to directly induce expression of these markers, suggesting that their expression is not a result of cell-autonomous deletion of caspase 8 but related to secondary effects caused by caspase 8 deletion in vivo (Supplementary Fig. 6d–g). Increasing evidence suggests that cancer cells, when undergoing an epithelial-to-mesenchymal-transition, transiently dedifferentiate and acquire stem-cell-like properties[39]. Notably, this process can be promoted by cancer-associated inflammation[40]. Intriguingly, loss of caspase 8 is capable of triggering an inflammatory response in healthy skin and ileum[41,42]. Given that deletion of Rb1 and Trp53 is known to

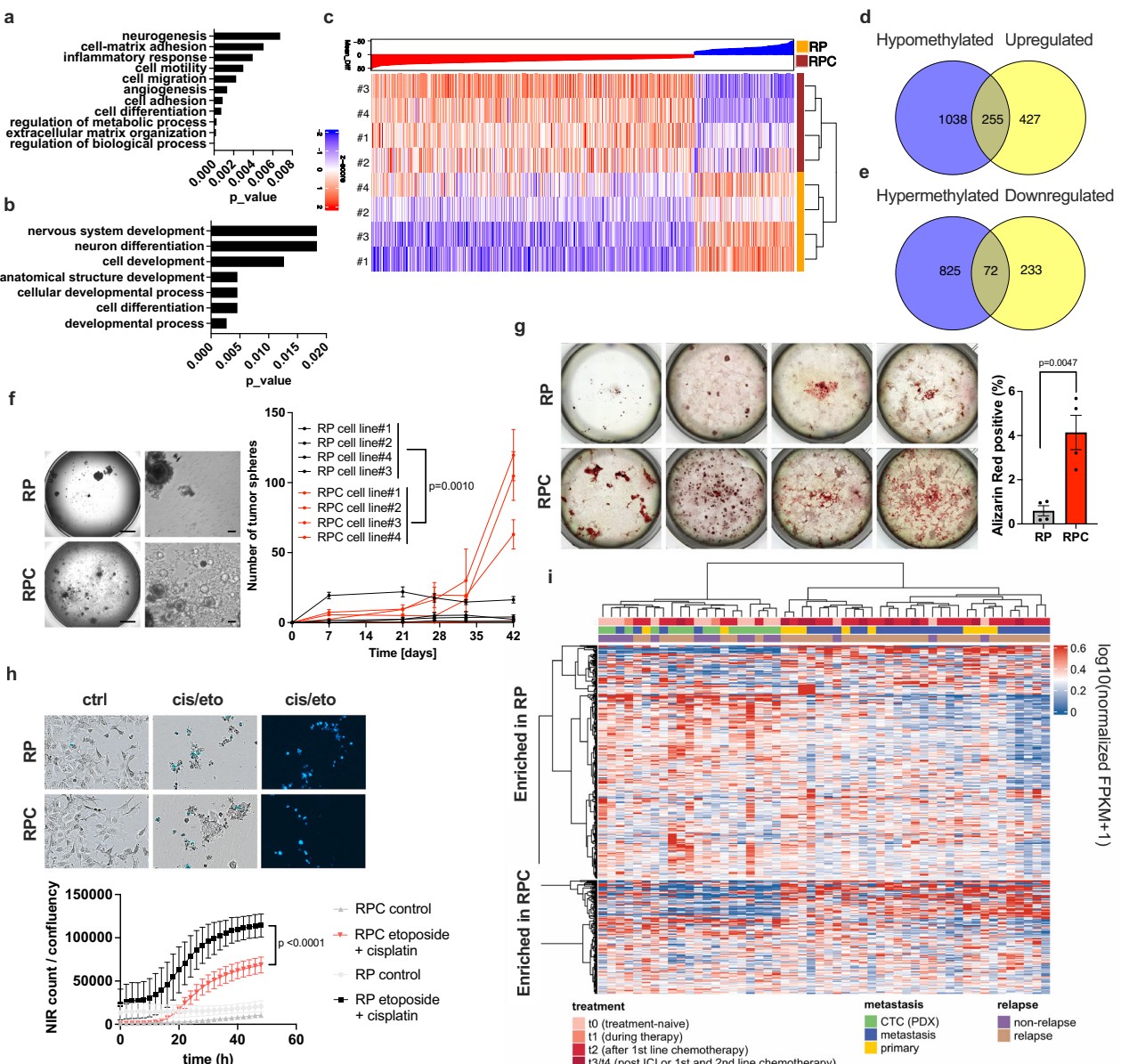

**Fig. 3 | Absence of caspase 8 promotes epigenetic reprogramming towards a stem cell-like state and phenocopies transcriptional profiles of human metastatic and relapse SCLC. a** Enriched pathways within the top 50 upregulated and **b** downregulated genes in RPC-derived cell lines. **c** RPC-derived cell lines (*n* = 4) as compared to RP-derived cell lines (n = 4) were subjected to MeDip-Seq. Differentially methylated regions (DMRs) within promotor regions between the 4 RP and 4 RPC cell lines are shown. Blue and red colors show increased hypomethylation and hypermethylation respectively, in RPC cell lines. **d** Venn diagrams of the overlap between significantly hypomethylated and significantly upregulated Gene Ontology terms in RPC cells (*p* = 2.68e−08) and **e** between significantly hypermethylated and significantly downregulated Gene Ontology terms in RPC cells (*p* = 2.54e−23). **f** RPC-derived cell lines (*n* = 4) as compared to RP-derived cell lines (*n* = 4) were subjected to spheroid assays for the indicated time. Data are means ± SEM of three replicates per cell line per time point of a representative experiment.

Representative images of RP (upper) or RPC (lower) tumor spheroids at the last time point. Scale bars: images on the left: 1000 μm, images on the right: 100 μm. **g** RP (*n* = 4) and RPC (*n* = 4) cell lines were cultured for 3 weeks in osteocyte differentiation media and subsequently stained with Alizarin Red. Data are presented as mean values of positive cells per well ± SEM. **h** RP- (*n* = 3) and RPC-derived cell lines (*n* = 3) were treated with cisplatin [20 μM]/ etoposide [10 μM], DRAQ7 was added to each well [100 nM] and live cell imaging (IncuCyte) was used to quantify dead cells (DRAQ7⁺) normalized to confluency. Data are means ± SEM of 3 biological replicates in 3 different cell lines per genotype. **i** Human SCLC patient RNA-seq data[20] (*n* = 52) were analyzed and clustered by transcriptional profiles up- and down-regulated in RPC versus RP mice from snRNA-seq pseudo-bulk. **a**, **b** Benjamini-Hochberg false discovery rate (FDR) correction was used to adjust for multiple testing. **d**, **e** Fisher´s exact test **f**–**h** two-tailed unpaired *t* tests. Source Data are provided as a Source Data file.

have a latency of 6–9 months before lung tumors can be detected in this model[23], we hypothesized that the immediate effect of caspase 8 deletion in pre-tumoral RPC lungs might be to fuel an inflammatory response in the lung prior to tumor development. Indeed, shortly after Rb1, Trp53 and Casp8 deletion, a significant amount of peri-bronchial CD45⁺ immune infiltration could be detected in RPC lungs (Fig. 4a and Supplementary Fig. 7a). Moreover, inflammatory cytokines and

chemokines consistent with a pro-inflammatory immune response including Cxcl1, IFN-γ and IFN-stimulated genes like Zbp1, as well as iNOS, were significantly upregulated 2 weeks after ad-Cre inhalation in RPC as compared to RP lungs (Fig. 4b). Other inflammatory genes such as Tnf, Il6, Il10, Cxcl2 and Ifnb also showed a trend towards higher expression levels in the lungs of RPC mice at this time point. Of note, inhaled C8^FL/FL lungs also showed peri-bronchial immune infiltration

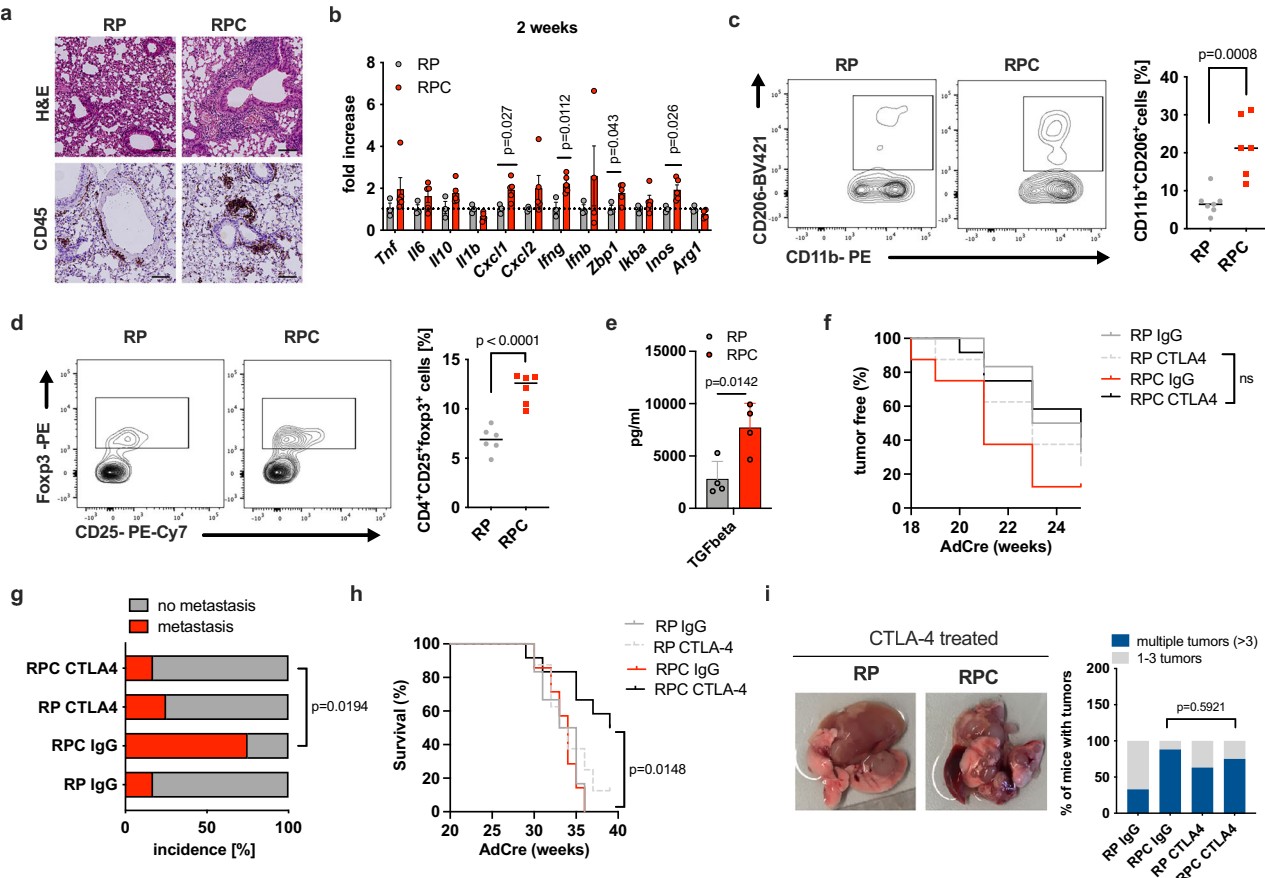

**Fig. 4 | Lack of caspase 8 expression initiates early immunosuppressive inflammation which promotes metastasis. a** Representative images of histological analysis (H&E) and immunostaining (CD45) of lungs of RP (*n* = 4) and RPC (*n* = 7) mice 2 weeks post Ad-Cre inhalation. Scale bars represent 100 μm. **b** qPCR analysis of whole lungs 2 weeks post Ad-Cre inhalation (RP *n* = 3 and RPC *n* = 5). RPC expression data were normalized to the housekeeping gene actin and plotted as fold increase compared to RP. Data are presented as mean values ± SEM. 12 weeks post Ad-Cre inhalation, immune cells were isolated from whole lungs of RP (*n* = 7) and RPC (*n* = 6) mice followed by FACS analysis. Proportions of **c** M2 macrophages (CD11b⁺CD206⁺) **d** Tregs (CD4⁺CD25⁺foxp3⁺) within CD45⁺ cells. **e** TGF-beta ELISA of protein extracts from whole lungs of RP (*n* = 4) and RPC (*n* = 4) mice 18 weeks post Ad-Cre inhalation. Data are presented as mean values ± SEM. **f** 12 weeks post Ad-Cre inhalation, RP and RPC mice were treated either with IgG [5 mg/kg] or anti-CTLA-4 [5 mg/kg] every other day for two weeks. RP IgG (*n* = 6), RPC IgG (*n* = 8), RP anti-CTLA-4 (*n* = 8) and RPC anti-CTLA-4 (*n* = 12). Tumor incidence was determined by time of first detection in MRI imaging until humane endpoint. **g** Metastasis incidence based on macroscopical analysis at humane endpoint. **h** Kaplan–Meier survival of the indicated genotypes after IgG or anti-CTLA-4 treatment. **i** Treated mice were sacrificed at humane endpoint. Representative photos and quantification of number of macroscopic nodules per lung of mice are shown. **b**–**e** Two-tailed unpaired *t* tests. **g**, **i** Fisher´s exact test (two-sided), **h**, **f** Mantel-Cox test. Source Data are provided as a Source Data file.

and increased pro-inflammatory genes, suggesting that this an effect of tissue damage caused by caspase 8 loss and not by Rb1 and Trp53 loss (Supplementary Fig. 7b, c). Interestingly, at this time point, in both RP and RPC pre-tumoral lungs, we also observed increased proportions of CD4⁺ T regulatory cells (CD25⁺FOXP3⁺, Tregs) and M2-macrophages (CD11b⁺CD206⁺) in comparison to non- inhaled lungs, indicating onset of immunosuppression that typically accompanies tumorigenesis (Supplementary Fig. 7d, e). Despite the fact that there was no significant difference in Treg and M2 macrophage percentages at this early time point between RP and RPC pre-tumoral lungs, FACS analysis at a later time point (12 weeks post inhalation, still pre-tumoral) revealed significantly increased proportions of Tregs and M2 macrophages in RPC lungs (Fig. 4c, d). Importantly, at this time point, the increase in proportions of M2 macrophages had almost resolved for RP but not for RPC lungs indicating persistence of immunosuppression in RPC mice (Supplementary Fig. 7f). The checkpoint receptors TIM3 and PD1 were also found to be significantly upregulated in RPC lungs (Supplementary Fig. 7g), despite no difference in the proportion of total CD45⁺ cells or other immune cell populations such as CD3⁺ or CD11b⁺ cells in RPC lungs (Supplementary Fig. 7h, i). Interestingly, RPC pre-tumoral lungs also showed a significant increase in

the proportions of B cells (CD19⁺) (Supplementary Fig. 7i). At the same time, we observed that proportions of naïve CD44⁻CD62L⁺ T cells were increased in RPC lungs while proportions of activated CD8⁺CD69⁺ T cells and effector Th17 cells (CD4⁺RORγ⁺) were decreased (Supplementary Fig. 7j–l). Notably, caspase 8 deletion within normal lungs did not phenocopy this effect (Supplementary Fig. 7m–r). Moreover, pre-tumoral lungs from RPC mice showed a significant downregulation of the expression of inflammatory cytokines like Tnf, Inos and Zbp1 compared to RP lungs at this time point (Supplementary Fig. 8a), while caspase 8 deletion alone in normal lungs did not show any appreciable differences (Supplementary Fig. 8b).

These results suggest that the initial inflammatory response induced by caspase 8 deletion observed at 2 weeks does not result in long-term inflammatory effects in normal lungs, while caspase 8 deletion in the context of Rb1/Trp53 co-deletion leads to persistence of immunosuppressive inflammation. In line with this, soluble levels of the immunosuppressive cytokine TGF-ß at an even later time point (18 weeks post inhalation) were significantly elevated in pre-tumoral RPC lung extracts (Fig. 4e), while this was not the case for pro-inflammatory cytokines like TNF, IL-6 or IL-10 (Supplementary Fig. 8c).

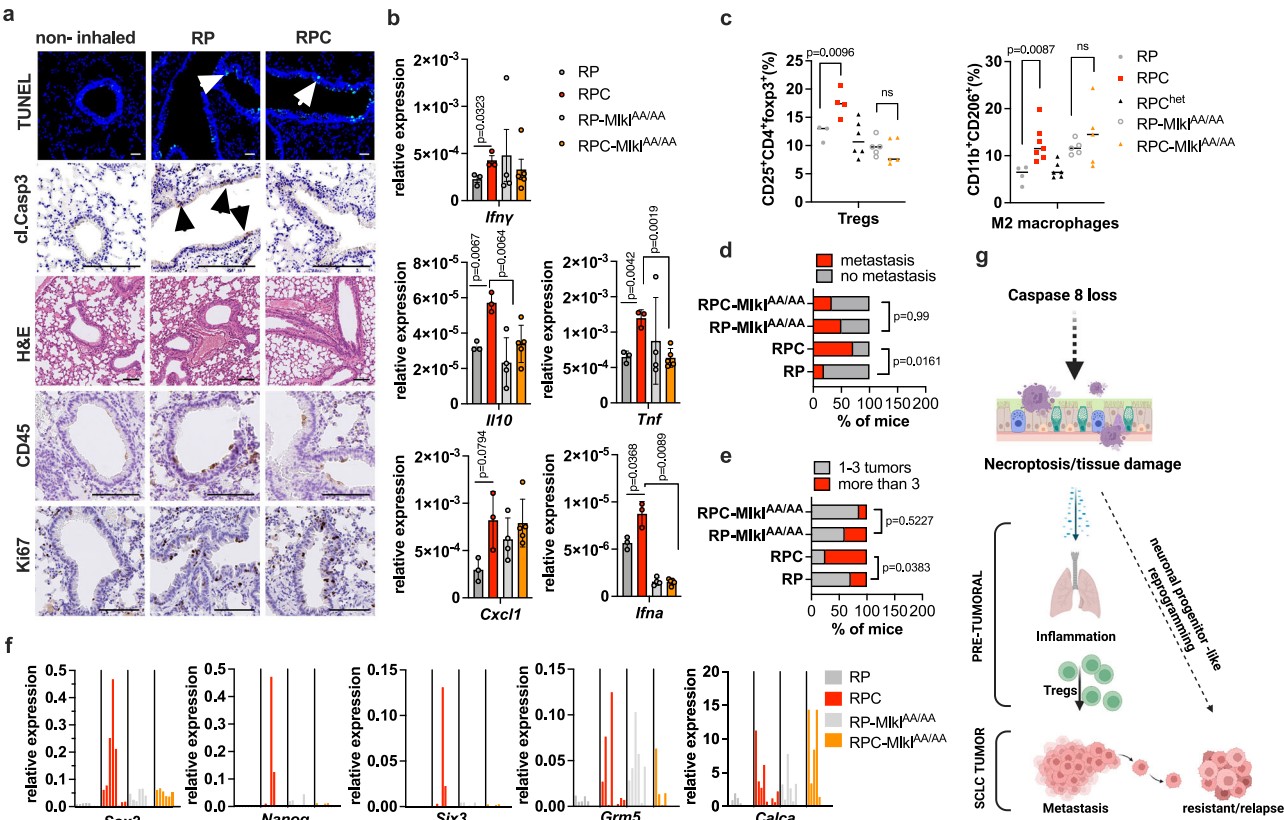

**Fig. 5 | Necroptosis promotes the establishment of immune-suppressive inflammation, metastasis and features of stemness. a** RP (n = 3) and RPC (n = 3) mice were sacrificed 5 days after Ad-Cre inhalation. Representative images of histological analysis and indicated stainings of lungs are shown. Non-inhaled controls received intratracheal PBS. Scale bars represent 10 mm for TUNEL stainings and 100 μm for all other. **b** qPCR analysis for the indicated genes from whole lungs 5 days post Ad-Cre inhalation are shown. RP (n = 3), RPC (n = 3), RP-Mlkl^{AA/AA} (n = 4) and RPC-Mlkl^{AA/AA} (n = 5). Data are presented as mean values ± SEM of the biological replicates **c** 12 weeks post Ad-Cre inhalation, immune cells were isolated from whole lungs of RP (n = 4), RPC (n = 6), RP-Mlkl^{AA/AA} (n = 5) and RPC-Mlkl^{AA/AA} (n = 5) mice followed by FACS analysis. Proportions of M2 macrophages (CD11b⁺CD206⁺)

and Tregs (CD4⁺CD25⁺foxp3⁺) within single/live immune cells are shown. **d** % of mice with or without metastasis 30 weeks post Ad-Cre inhalation. RP (n = 26), RPC (n = 7), RP-Mlkl^{AA/AA} (n = 6) and RPC-Mlkl^{AA/AA} (n = 6). **e** % of mice with more or less than 3 macroscopic lung tumor nodules 30 weeks post Ad-Cre inhalation, RP (n = 24), RPC (n = 8), RP-Mlkl^{AA/AA} (n = 5) and RPC-Mlkl^{AA/AA} (n = 7). **f** qPCR analysis comparing gene expression between RP (n = 5), RPC (n = 9), RP-Mlkl^{AA/AA} (n = 6) and RPC-Mlkl^{AA/AA} (n = 6) endpoint tumors. Relative expression compared to the housekeeping gene actin is shown. **g** proposed sequence of events in the RPC mouse. Created in BioRender. Stroh, J. (2025) https://BioRender.com/k47q488. **b** Two-tailed unpaired t tests. **c** One-way ANOVA (uncorrected Fisher´s LSD). **d**, **e** Fischer´s exact test. Source Data are provided as a Source Data file.

Tregs exhibit potent effector cell inhibitory activity in various neoplastic diseases and have been linked with increased metastatic burden[43]. Given that we found that pre-tumoral caspase 8 deletion in the context of Rb1/Trp53 co-deletion led to increased proportions of Tregs, we hypothesized that Tregs might be responsible for the increase in metastatic disease observed in endpoint RPC mice. To test this, we made use of the fact that anti cytotoxic T-lymphocyte associated protein 4 (CTLA-4) antibodies can functionally increase the ratio of effector T cells (Teff) to Tregs[44]. Treatment of RP and RPC mice with CTLA-4 blocking antibodies (12 weeks post inhalation, pre-tumoral) efficiently reduced proportions of Tregs and simultaneously induced proportions of effector CD8⁺ T cells in pre-tumoral lungs, resulting in a significant increase in the Teff/Treg ratio in both RP and RPC lungs (Supplementary Fig. 8d). Importantly, this early treatment translated to subsequent neutralization of accelerated tumor development, reversal of increase in metastasis and significantly increased survival in RPC mice (Fig. 4f–h).

At experimental endpoint, around 30 weeks post inhalation, RPC lungs still presented with no differences in pro-inflammatory gene expression, but rather a trend towards lower Ifny and Ifnb expression suggesting that the pre-tumoral immunosuppressive inflammatory environment persists until endpoint (Supplementary Fig. 8e). Notably, Treg inactivation did not reverse multiplicity of tumor formation in

RPC mice (Fig. 4i) nor did it affect increased expression levels of neuronal progenitor-like genes in endpoint RPC tumors (Supplementary Fig. 8f). These data suggest that while enhanced Treg-mediated inflammation is sufficient to promote metastatic disease of RPC mice, it is dispensable for the emergence of a neuronal progenitor cell-like state that we observed in RPC tumors and found enriched in human SCLC relapse samples. Vice versa, the data indicate that the neuronal progenitor cell state without Treg help is insufficient to promote metastatic disease.

## A Necroptosis-prone tumor niche promotes the establishment of immune-suppressive inflammation, metastasis and features of a neuronal progenitor-like state

Lack of caspase 8 allows for induction of necroptosis[10,45], a type of regulated necrosis resulting in plasma membrane rupture via MLKL[11,12] and inflammation[46]. Based on this, we hypothesized that the increased pre-tumoral immune-suppressive inflammation we observed in RPC mice might derive from necroptosis-fueled inflammation. Indeed, Ad-Cre inhalation induced a rapid cell death event 5 days after inhalation in lung bronchial cells as seen by positive TUNEL stainings in both RP and RPC mice. Importantly, in RP lungs this cell death was also positive for cleaved caspase 3 and hence apoptotic, while cleaved caspase 3 could not be detected in RPC lungs (Fig. 5a). In support of tissue injury

and remodeling, histological analysis showed compensatory proliferation and presence of a few immune cells within the bronchial epithelium, but not yet a strong peri-bronchial immune infiltration as it was observed in the 2-week post inhalation time point (Fig. 5a). These data suggested that while cell death was evident in both cases, upon caspase 8 deletion apoptosis was switched to necroptosis which might lead to the tissue inflammation observed from 2 weeks after induction onwards given the inflammatory nature of necroptosis in comparison to apoptosis. Interestingly, in vitro deletion of Rb1 and Trp53 in primary mouse lung fibroblasts (PMLFs) isolated from non-inhaled RP mice showed induction of γ−H2AX phosphorylation, indicative of DNA damage, and phosphorylation of MLKL, suggesting induction of necroptosis. Importantly, in PMLFs from RPC mice, despite no additional induction of DNA damage, we could show significantly enhanced MLKL phosphorylation which was reverted by nec1s co-treatment indicating RIPK1-dependent induction of MLKL phosphorylation as characteristic for necroptosis (Supplementary Fig. 9c). These data suggest that necroptosis may be triggered by the loss of Rb1 and Trp53 and is significantly enhanced in cells that additionally lack caspase 8. To test this hypothesis genetically, we crossed RPC mice to RP-Mlkl$^{AA/AA}$ mice[47] in which MLKL cannot be phosphorylated by receptor-interacting protein receptor 3 (RIPK3) anymore and, hence, is incapable of executing necroptotic cell death (RPC-Mlkl$^{AA/AA}$) mice. Indeed, elevated pro-inflammatory chemo- and cytokines present in RPC mice were neutralized within the Mlkl$^{AA/AA}$ genetic background (Fig. 5b). Notably, cytokine levels of normal lungs with caspase 8 deletion were similar to the levels observed in RPC lungs, yet additional MLKL inactivation (Mlkl$^{AA/AA}$;C8$^{FL/FL}$) did not reverse these levels in normal lungs as previously reported[48,49] (Supplementary Fig. 9b). Importantly, the induction of M2 macrophages and Tregs caused by lack of caspase 8 was also neutralized upon concomitant MLKL inactivation (Fig. 5c) suggesting that MLKL-induced inflammation is responsible for their increased presence in RPC mice. As expected, MLKL inactivation additionally restored levels of B cells and effector cells in the pre-tumoral lungs (Supplementary Fig. 9c−f).

To determine whether reduced apoptosis might also be responsible for the effects observed we compared RPC mice heterozygous for caspase 8 (RPC$^{het}$) with their RP and RPC littermates to distinguish between the effect of necroptosis induction and lack of apoptosis due to caspase 8 deletion. Interestingly, RPC$^{het}$ mice reversed the increase in cytokines and chemokines and the peri-bronchial inflammation observed at 2 weeks post inhalation in RPC mice (Supplementary Fig. 10a, b). Moreover, RPC$^{het}$ mice also showed levels of Tregs and M2 macrophages comparable to RP mice (Fig. 5c), supporting the finding that necroptosis and not loss of apoptosis drives pre-tumoral inflammation in RPC mice. Accordingly, the increased metastatic incidence and multiplicity of primary tumor nodule formation in RPC mice was almost completely reversed upon functional inactivation of MLKL (Fig. 5d, e, and Supplementary Fig. 10c, d). Additionally, in bulk qPCR analysis of individual endpoint tumors we observed that the upregulation of genes associated with the neuronal progenitor-like state in RPC tumors was also reversed upon genetic inactivation of necroptosis in RPC-Mlkl$^{AA/AA}$ tumors (Fig. 5f). Notably, Calca expression was also increased in RPC tumors, but this was not changed upon MLKL removal (Fig. 5f). Given the fact that bulk expression does not exclude signals from non-tumor cells and tumor purity varied between samples, we also checked cell lines that we generated from RP-Mlkl$^{AA/AA}$ and RPC-Mlkl$^{AA/AA}$ tumors. Immunoblot analysis confirmed absent expression of caspase 8 in RPC-Mlkl$^{AA/AA}$ cell lines (Supplementary Fig. 10e). In this case, similarly to bulk endpoint tumor expression patterns, Nanog, Six3, Dcx, but interestingly, also increased Calca expression, was reversed (Supplementary Fig. 10f). Importantly, cell lines from RP-Mlkl$^{AA/AA}$ and RPC-Mlkl$^{AA/AA}$ tumors did not show any differences in their ability to differentiate into adipocyte-like cells (Supplementary

Fig. 10g), further indicating loss of pluripotent features in RPC-Mlkl$^{AA/AA}$ cells.

Together, these data place necroptosis upstream of both increased immune-suppressive inflammation and metastasis as well as emergence of a neuronal progenitor cell-like state in SCLC (Fig. 5g). In support of the notion that the RP-mouse model of SCLC does not sufficiently mimic absent caspase 8 expression and, hence, does not spontaneously allow for necroptosis-associated inflammation, RP-Mlkl$^{AA/AA}$ mice did not show a significant difference in tumor incidence and Kaplan−Meier survival nor showed any changes in tumor-associated immune infiltration (Supplementary Fig. 11a−d). Collectively, our data support the concept wherein pre-tumoral necroptosis promotes the presence of a neuronal progenitor-like cell state associated with relapse SCLC along with an early immunosuppressive niche which in turn promotes metastatic disease in SCLC.

## Discussion

Here, by generating an SCLC mouse model (RPC) that phenocopies absent protein expression of caspase 8 as seen in most human patients, we have identified a pathological role for lack of caspase 8 expression in promoting SCLC. Notably, RPC tumors retain NE identity but are characterized by a neuronal progenitor-like reprogramming that is accompanied by features of plasticity, as well as increased metastasis incidence. Mechanistically, we show that necroptosis and pre-tumoral inflammation induced by lack of caspase 8 is sufficient to induce these malignant features.

In recent years, tumor plasticity has been recognized as a new hallmark of cancer[50]. Our results show a remarkable loss of neuronal differentiation in RPC tumors with concomitant gain of stem cell-like features, suggesting that neuronal progenitor mimicry is a previously unrecognized feature of SCLC. Indeed, Sox2, a key driver of a stem cell fate[51], was one of the factors significantly upregulated in RPC tumors. Notably, RPC tumor cells showed significantly altered methylation profiles suggestive of loss of neuronal differentiation at epigenetic level although further studies are required to functionally assess this correlation.

The neuronal progenitor cell-like state described in our study does not follow the NE/non-NE plasticity seen in MYC-driven SCLC[34], but remains well within a spectrum of NE differentiation. Human NE transcriptional subtype variants with profound similarities to RPC tumors have been described. The NEv2 or SCLC-A2 was shown to be specifically enriched in liver metastases and is thought to represent a highly plastic cell state that possibly associates with drug resistance and poor prognosis[31,32]. Similar to RPC tumors, human NEv2 SCLC was found to be enriched in Calca transcripts, while it also presented with a profound lack of neuronal pathway signatures compared to the other transcriptional NE subtypes, SCLC-A and -N[32]. A similar NE variant within the SCLC-A ASCL1-positive subtype, ASCLC-Aσ, was identified in human and mouse SCLC metastases samples and also presented with loss of neuronal identity, controlled by Sox1 and Nkx2-1 (also known as TTF-1)[52]. Interestingly, RPC tumors also showed a significant down-regulation of Sox1, as well as lower levels of Nkx2-1. In further support of the existence of stem-cell-like ASCL1$^+$ NE cancers, ASCL1 was shown to activate an epigenetic neuronal stem cell-like reprogramming in prostate cancer[53].

Caspase 8 acts as a key regulator that promotes apoptosis and prevents necroptosis. While necroptosis can both promote and inhibit tumor growth in other cancers, in our model, it occurs within the pre-tumoral niche. This early induction may allow inflammation to stabilize into an immune-suppressive state before cancer cells appear, favoring a tumor-supportive role of necroptosis, as no tumor cells are directly killed and only the resulting inflammation influences tumor growth. Interestingly, chronic inflammation in the lung often triggered by infections is a well-established risk factor for lung cancers[54,55] and

particularly patients with chronic obstructive pulmonary disease have a significantly higher risk of lung cancer[56]. Moreover, inflammation has been linked with the emergence of cancer cell dedifferentiation and epigenetic plasticity in pancreatic cancer[57]. In cases of necroptosis induction in established tumors besides the tumor cell loss, necroptosis was suggested to provide an ample supply of tumor-associated antigens to dendritic cells, culminating in an anti-tumor response as observed in necroptosis-inducing tumor vaccination experiments[13]. It is therefore possible that the time at which necroptosis is induced along the tumor development axis may determine its impact on tumor development.

We show that loss of caspase 8 triggers an early inflammatory response that evolves into an immunosuppressive environment promoting metastasis. In RPC mice, the rise in metastasis is fully reversed by MLKL inactivation, indicating that necroptosis-driven inflammation—particularly enhanced Treg presence—supports metastasis. Early Treg removal abolishes the rise in metastasis observed in RPC mice, and genetic necroptosis inhibition prevents enhanced Treg presence. However, Treg removal does not affect stem-like marker expression in RPC cancer cells, suggesting that the neuronal progenitor-like state alone does not drive metastasis, though it may contribute to Treg recruitment. Thus, while we identify neuronal progenitor mimicry as a feature of SCLC, its role in metastasis requires further functional studies. Notably, necroptosis inhibition reverses neuronal progenitor markers, but Treg inactivation does not. These results imply that the neuronal progenitor-like state arises from inflammation preceding enhanced Treg induction or as a result of a cell-autonomous effect of caspase 8 loss. Yet, ex vivo caspase 8 silencing in RP-derived SCLC cells fails to induce these genes, suggesting the state results from inflammation-driven, possibly epigenetic, changes.

Although MLKL loss in human SCLC suggests selection against necroptosis[17], whether similar pre-tumoral inflammation occurs in patients remains unknown due to late diagnosis and biopsy. Caspase 8 protein expression has previously been shown to be very low-to-absent in human SCLC cell lines due to epigenetic silencing of the caspase 8 promotor region[7,58,59]. As a functional consequence, human SCLC cell lines were shown to be entirely resistant to extrinsic apoptosis[6,17]. While the RPC mouse model does not recapitulate epigenetic loss of caspase 8 and, hence, cannot model phenotypes caused by epigenetic regulation, it does mimic absence of caspase 8 protein expression as observed in 80% of SCLC cell lines[60]. Importantly, functional heterozygosity of the caspase 8 locus in vivo still allows for extrinsic apoptosis to proceed[9], indicating that heterozygous caspase 8 deletion in RP mice would not functionally mimic the situation found in human SCLC cell lines. Together, these data suggest that homozygous deletion of caspase 8 represents the best approximation of the lack of caspase 8 protein expression in human SCLC.

Notably, we observed a more prominent reduction of caspase 8 protein levels than mRNA in human SCLC samples. This may suggest translational or post-translational regulation in addition to epigenetic silencing. Indeed, it is interesting to note that caspase 8 stability was shown to be promoted by cullin 3-mediated K63 polyubiquitination[61], and neuroendocrine cancers seem to overexpress cullin 3[62]. However, since the human datasets are derived from biopsy samples containing both tumor cells and non-transformed microenvironmental cells, we cannot determine which specific cell type is responsible for this discrepancy.

Based upon these collective data, we propose that in SCLC and possibly other NE cancers, lack of caspase 8, does not only confer apoptosis evasion, but may promote SCLC aggressiveness via enabling necroptosis-fueled inflammation. Taken together, our data uncover an unexpected role for lack of caspase 8 expression in directing neuroendocrine plasticity towards a neuronal precursor cell-like state, increased immunosuppressive inflammation and metastasis.

## Methods

### Mice
Rb[FL/FL] mice[63] and Tp53[FL/FL] mice[64] have been described previously and were provided by the lab of H. Christian Reinhardt. Casp8[FL/FL] mice[25] on a C57BL/6 background were obtained under a material transfer agreement from Stephen Hedrick. Mlkl[AA/AA] mice were newly generated in the Pasparakis lab and described in Körner et al.[47]. Animals were maintained at the SPF animal facility of the CECAD Research Center at the University of Cologne on a 12-h light/dark cycle with water and food ad libitum. Mice that were monitored by MRI for tumor development were transferred and maintained until endpoint to the Nuclear Medicine Animal Facility of the University Clinic of Cologne. All animal experiments were approved by local government authorities (Landesamt für Verbraucherschutz und Ernährung, Nordrhein-Westfalen, Germany) and were conducted in compliance with European, national and institutional guidelines on animal welfare.

### Clinical samples
SCLC patient specimens were described before[20], and the subset analyzed for DNA methylation in this study is listed in Supplementary Data 1. The study was approved by the institutional ethics review board of the University of Cologne. Written consent was obtained from all patients included in the study.

### Inhalations/MRI
For induction of lung tumor formation, 8–12 weeks old mice were anesthetized with Ketavet (100 mg/kg) and Rompun (20 mg/kg) by intraperitoneal injection followed by intratracheal inhalation of replication-deficient adenovirus expressing Cre (Ad5-CMV-Cre, $2.5 \times 10^7$ PFU, University of Iowa) mixed with OptiMEM and CaCl2 (10 mM). 3 months after inhalations, tumor formation was monitored bi-weekly by MRI using A 3.0T Philips Achieva clinical MRI (Philips Best, the Netherlands) attached to a dedicated mouse solenoid coil (Philips Hamburg, Germany). T2-weighted MR images were acquired in the axial plane using turbo-spin echo (TSE) sequence [repetition time (TR) = 3819 ms, echo time (TE) = 60 ms, field of view (FOV) $=40 \times 40 \times 20$ mm$^3$, reconstructed voxel size$=0.13 \times 0.13 \times 1.0$ mm$^3$, number of average=1) under isoflurane (2.5%) anesthesia. MR images (DICOM files) were analyzed by determining and calculating region of interests using Horos v3.3.1 software. Mice were sacrificed at indicated experimental endpoint (time) or later at humane endpoints when reaching a score 10 (moderate burden) within a scale of 20 (severe burden) to minimize animal suffering.

### CTLA-4 treatment
2 groups of mice (one with males and one with females only) were inhaled with Ad-Cre as described above. At week 12 post Ad-Cre inhalation mice were treated (i.p.) with anti-CTLA-4 depletion antibody (anti-CTLA-4, Clone UC10-4F10-11, InVivoPlus, BioXcell) or the respective IgG (polyclonal Armenian Hamster IgG, InVivoPlus, BioXcell) every other day, for 2 weeks (5 mg/kg in 200 µl PBS). Mice were monitored bi-weekly by MRI until they were sacrificed according to endpoint humane criteria.

### Cell lines
Human SCLC cell lines (H889, H526 and H82) as well as human NSCLC cell lines (H460, H441 and A549) were obtained from ATCC and were cultured in RPMI medium + 1% Pen/Strep + 10% FCS and grown as suspension or adherent cells, respectively. Mouse SCLC cell lines (AVR 428.1, AVR 404.1G, AVR 132.2) as well as mouse NSCLC cell lines (ACF 132.2, ACF 1035.2 and ACF 135.1) were previously derived from lung tumors of RP-mice (SCLC) or KP-mice (NSCLC) and cultured in RPMI medium + 1% Pen/Strep + 10% FCS as adherent cell lines. Mouse SCLC cell lines RP181.5, RP252.7, and RP285.5 were kindly provided from H. Christian Reinhardt. Cas9-expressing RP cell lines were previously

generated in the lab and reported[17]. For generation of tumor-cell lines from RP, RPC, RP-Mlkl$^{AA/AA}$ and RPC-Mlkl$^{AA/AA}$ mice, fresh tumors were isolated at the humane endpoint. The tumors were incubated in 2.5% Trypsin 10X (Gibco) at 37 °C and 5% $CO_2$ for 20 min. RPMI 1640 medium containing 10% FCS and 1% P/S was added, and the tissue was incubated at 37 °C, 5% $CO_2$ overnight. The next day, the remaining tumor tissue was removed from the culture, and the cells were kept at 37 °C with 5% $CO_2$ until they grew confluent. All cells were tested for mycoplasma at regular intervals (mycoplasma barcodes, Eurofins Genomics), and human cell lines have been validated by short tandem repeat (STR) profiling provided by Eurofins Genomics.

### Primary lung fibroblasts generation and in vitro transduction with AdCre

Isolated whole lungs were digested in 3 mg/ml Collagenase Type II (Worthington LS004176) Solution in DMEM w/o serum (DMEM+Na-pyruvat+P/S+Glutamin) for 45 min at 37 °C. Solution was homogenized with long canula by slowly pipetting several times and plated onto 3 culture dishes (15 cm Ø) contain warm complete medium. The next day, supernatants were collected, washed 1× with PBS, and added back into the plates. Cells were let to grow until confluency and used for experiments or stored by freezing. For in vitro transfection with adeno-Cre, cells were seeded at 100,000/6-well. The next day FCS-containing medium was removed, cells were washed once with PBS, and 1 ml FCS-free medium was added for 2 h prior to the transfection. 250 µl Opti-MEM + 0.75 µl Polybrene +2 µl Adeno-Cre virus (300pfu/cell) was added to the cells for 24 h. The next day media were removed, cells were washed and incubated for 48h before analyzed.

### siRNA transfection

For the knockdown of Caspase 8 using L-043044-00-0005 ON-TARGETplus Mouse Casp8 siRNA, 200 µL Opti-MEM were mixed with 1.5 µL Dharmafect Reagent I per single 6-Well and incubated 10 min at room temperature. 2.2 µL siRNA (Stock 20 mM) per 200 µL Opti-MEM mixture were added and incubated for another 30 min at room temperature. 200 µL of the mixture were added to each well of a 6-well plate, 150,000 cells were plated on top in 1 mL media and incubated for 48 h at 37 °C.

### TrueGuide sgRNA transfection

For the knockout of Caspase 8 with A35533 TrueGuide™ Synthetic sgRNA Casp8, 200,000 cells were plated 1 day prior in a 6-well plate in RPMI + 10%FCS. For the transfection, 125 µL Opti-MEM were mixed with 37,5 pmol sgRNA (Stock 100 µM) in one tube and in another tube 125 µL Opti-MEM and 5 µL Lipofectamin 2000 were mixed per single 6-well. After 5 min of incubation at room temperature, the mixtures were combined and incubated for another 20 min at room temperature. 250 µL of the mixture were then added drop-wise to the cells. After 6 h transfection, cell culture medium was replaced with RPMI 10% FCS 1%P/S and incubated for 48 h at 37 °C.

### Tumor sphere formation

Tumor sphere formation was assessed by plating RP and RPC cell lines in triplicates at low densities (250–500 cells per 100 µl) in ultra-low attachment 96-well plates (Corning). Brightfield images of all wells were obtained every week using a Keyence BZ-X800 Microscope. Tumor spheres were counted at every time point, and the mean of each triplicate was calculated for every cell line per time point.

### Osteocyte differentiation assay

Cells were seeded at $3 \times 10^3$ cells/well in 12-well plates in 1 ml/well of differentiation media (osteocyte differentiation STEMPRO® Osteogenesis Differentiation Kit, Thermo Fisher, A1007201) or RPMI with 10% FCS plus 1% P/S media (for control). The medium was refreshed every 3 days and kept overall for 21-days. Then cells were fixed with 4%

paraformaldehyde and stained with 2% Alizarin red S solution (Merck, TMS-008-C) according to manufacture instructions. The pictures were acquired using the Keyence BZ-X800 microscope.

### Adipocyte differentiation assay

The cells were seeded at $12 \times 10^3$ cells/well in 12-well plates in 1 ml/well of RPMI with 10% FCS plus 1%P/S media. In 2 days, medium was replaced to adipocyte differentiation medium (Thermo fisher, A1007001). The medium was refreshed every 3 days and kept overall for 21-days. Then cells were fixed with 4% paraformaldehyde and stain with Oil Red O (Sigma-Aldrich, 1024190250) according to manufacture instructions. The pictures were acquired using the Keyence BZ-X800 microscope.

### PCR/quantitative real-time PCR (qPCR)

For qPCR analysis, small pieces of tumor or normal lung tissue were collected in RNALater and stored in −20 °C until further processing. For RNA isolation, 30 mg of tissue were mixed with ceramic beads in 350 µl RNA Lysis buffer (NucleoSpin RNA isolation kit, Macherey Nagel Ref. 740955.250) and were homogenization in a peqlab vial for 2 ×30 s using the Precellys 24-dual homogenizer (Peqlab). RNA isolation from lung/tumor tissue was performed according to the manufacturer's protocol. The isolated RNA was reverse transcribed into cDNA using the LunaScript RT SuperMix Kit (E3010L, NEB) following the protocol provided by the manufacturer. For quantitative PCR, 5 µl of Power SYBR Green PCR Master Mix (4368702, Thermo Fisher) was mixed with 2 µl of nuclease-free water (NEB), 1 µl (10 µM) of forward and reverse primers (Supplementary Data 2) and 2 µl of cDNA (5 µg/µl). Real-time qPCR was performed using the Quant Studio5 qRT PCR machine. Results were normalized to the expression of the house-keeping gene Actin. Genomic DNA from generated RP and RPC cell lines was extracted according to standard protocols and genotyping PCR was performed for the validation of Rb1 and Tp53 deletion.

### Methylated DNA immunoprecipitation sequencing (MeDIP-seq)

DNA of matched tumor-normal samples from 33 SCLC patients[20] was subjected to methylated DNA immunoprecipitation using an antibody targeting methylated cytosines followed by massive parallel sequencing (MeDIP-seq). DNA was isolated from fresh frozen tissue as described[20] and quality was accessed on an agarose gel. For each sample, 1 µg of DNA was sheared to a size of 200 to 500 bp using the Bioruptor NGS (Diagenode, Seraing, Belgium). Library preparation and MeDIPs were performed using the automated system IP-Star SX-8G Compact (Diagenode) with the iDeal library preparation kit (C05010020, Diagenode), followed by purification without size selection by adding 86.5 µl AmpPure XP beads (A63881, Beckman Coulter) to 100 µl of ligation reaction and immunoprecipitation with the auto MeDIP kit (C02010011, Diagenode). Antibody incubation lasted for 15 h at 4 °C. Immunoprecipitated samples were purified with the Auto IPure kit v2 (C03010010, Diagenode), PCR amplified and sequenced at the Max-Planck Institute for Molecular Genetics on a HiSeq 2000 (Illumina, San Diego, USA).

### Bioinformatic analysis of MeDIP-seq

Sequencing reads were aligned to the reference genome version hg38 (Genome Reference Consortium GRCh38) using BWA v0.7.15 aln followed by samse modules[65]. Subsequently, aligned MeDIP-seq reads were processed in R with QSEA v.1.14.0[66], according to the package recommendations. The data was calibrated based on DNA methylation from GSE68379[67] and GSE145156[68]. We considered a 250 bp region to be differentially methylated (DMR) if the adjusted for multiple testing (false discovery rate) $p$-value was smaller than 0.001 with additional threshold of methylation data to be available for at least 30 samples. DMRs were annotated with BSgenome v.1.56.0. and RefSeq release 71, ENCODE Encyclopedia v.3 as of 24th April 2014.

## Methyl-seq

Mouse samples were processed for Methyl-seq. In brief, genomic DNA was extracted using the QIAamp DNA Mini Kit (Qiagen, Hilden, Germany). 3 µg DNA was sheared using a BioRuptor NGS (Diagenode, Seraing, Belgium) and subjected to library preparation using the SureSelectXT Methyl-seq library preparation kit (Agilent Technologies, Waldbronn, Germany) followed by target enrichment with the SureSelectXT Mouse Methyl-seq target enrichment panel (109 Mb, #931052, Agilent Technologies). Subsequently, enriched DNA was bisulfite-converted using the EZ DNA methylation kit (Zymo Research, Freiburg, Germany) and libraries were PCR-amplified according to the manufacturer's instructions. Sequencing was performed at the Cologne Center for Genomics, Cologne, Germany on a NovaSeq 2000 platform.

## Bioinformatic analysis of Methyl-seq data

Quality of the paired-end data were checked with FastQC v0.11.8 [https://www.bioinformatics.babraham.ac.uk/projects/fastqc]. Reads were mapped against mm39 (mm39, Genome Reference Consortium Mouse Build 39 (GCA_000001635.9)) using BSMAP v2.90. BSMAP was run with the default parameters, except for the maximum allowed mismatches (-v 0.1). BAM files were generated, sorted and indexed with Samtools v1.4. Samtools v1.4 was also used to generate the mapping statistics excluding secondary alignments (-F 256)[69]. liftOver was applied to the BED file for the target regions (mm9) twice (mm9 to mm10 and mm10 to mm39) to reach the final genome version[70]. The intersection of reads with the target regions of the enrichment panel was calculated using IntersectBed v2.30.0[71]. BAM files containing only the target regions were generated and used as input for the methratio.py post-mapping script in the BSMAP package. Methylation values were calculated for each base and context (CG, CHG and CHH) using the parameters (-u -p -g -x [CG|CHG|CHH])[72]. BedGraph formatted files were generated using chromosomal positions in combination with coverage or methylation ratios. Defiant was used to call differential methylated regions between the two groups[73]. The significant DMRs (adjusted $p$-value < 0.05) were annotated with known mm39 genes from UCSC using the ChIPseeker R package[74]. Promoter regions were defined as ±3 kbp from transcription start site. Gene set analysis was performed with gProfiler80 or with STRING (string-db.org).

## Single-nucleus RNA sequencing (snRNA-seq)

Single nuclei suspension was isolated from fresh-frozen RP and RPC endpoint lung tumors as described in the Ren lab ENCODE in situ HiC protocol for tissue with minor modification. In brief, 15–50 mg fresh-frozen lung tissue was minced on ice and dissociated in 3 ml Lysis BufferI[5 mM CaCl₂, 3 mM MgAc, 2 mM pH8.0 EDTA, 0.5 mM EGTA, 1 mM DTT, 0.0001 M PMSF, one tablet cOmplete PI EDTA-free in every 25 ml buffer, 0.04U/ml murine RNase inhibitor] in gentleMACS C-tubes (Miltenyi Biotec 130-093-237) using program "Protein_01_01" on the gentleMACS Octo Dissociator (Miltenyi Biotec 130-096-427). After dissociation, 3 ml Lysis Buffer II [0.4% Triton X-100 in Lysis BufferI] was added to terminate the digestion. Tissue debris was filtered out through a 40 µm cell strainer. Nuclei pellets were collected by centrifugation at $450 \times g$ for 5 min. After resuspension in Lysis Buffer III [Lysis BufferIand Lysis BufferII1:1], pellets were further purified by gradient centrifuge in Sucrose Buffer [1 M sucrose, 3 mM MgAc, 10 mM Tris·HCl] at $450 \times g$ for 5 min. Nuclei pellets were washed, resuspended in DPBS (with 0.5µ/ml murine RNase inhibitor) and counted under the microscope to ensure less than 4 million single nuclei in each sample. Two million fixed nuclei were used for barcoding and sequencing library preparation using the Evercode WT Mega v2 kit (Parse Biosciences, ECW02050) according to the V2.1.1 user manual. Briefly, fixed nuclei were thawed on ice, counted using an automated cell counter (Bio-Rad, TC20), and distributed into the Round 1 Plate for reverse transcription barcoding. Upon completion of the reverse transcription, nuclei from all wells were pooled, pelleted at $450 \times g$ for 10 min at

4 °C, and then distributed into Round 2 Plate, followed by Round 3 Plate. Barcoded single-nuclei were strained through a 15 µm cell strainer and evenly divided into sub-libraries of 62,500 nuclei each. After cDNA amplification and library purification, sequencing libraries were quantified with the Qubit dsDNA High Sensitivity kit (Invitrogen, Q32851) and separated on an Agilent Bioanalyzer using the High Sensitivity DNA Kit (Agilent, 5067-4626). Sequencing libraries were sequenced on an Illumina NovaSeq 6000 platform, resulting in an average depth of 129,496,631 reads/sample.

## Quality control and clustering analysis

The sequencing data in fastq format were aligned to the GRCm39.110 reference and demultiplexed into individual samples using the PARSE snRNAseq pipeline v.1.1.1 to generate count matrices for each sample. Count matrices of each sample were converted into Seurat objects. All data analysis was performed using Seurat (v4.4.0) following the official vignette. Low quality nuclei e.g., low number of counts (<300) and high mitochondria gene content (>10%) were removed. Possible multiplets with high number of genes (>12000) were also excluded from further analysis. After filtering, 8340 cells remained in RP1, 8848 cells in RP2, 10853 cells in RPC1 and 10161 cells in RPC2. RNA counts of each sample were normalized by SCTransform (v2) using the Seurat package. All four samples were merged into one Seurat object for Principle Component Analysis (RunPCA), clustering (FindClusters, BuildClusterTree) and UMAP projection (RunUMAP). The first 20 dimensions from PCA and 0.4 resolution was used for subsequent clustering analysis. To identify cancer cells, all four samples were integrated into an annotated healthy single-cell sequencing dataset (Tabula Muris microfluidic cells in lung and trachea). After clustering analysis, clusters mainly composed (>95%) of RP or RPC samples were identified as tumor cell clusters. Vice versa, query sample clusters clustering together with the normal reference dataset were identified as normal cells and removed from further tumor cell analysis. Annotation of nonmalignant cells within query samples was further manually confirmed by maker gene expression of each normal cell type (FindAllMarkers). Malignant cells were subset and downsampled to the same number of cells (7000/sample). Again, the first 15 dimensions and a resolution 0.4 were used for clustering of tumor cells.

## Pseudobulk differential expression analysis

To identify differentially expressed genes between RP and RPC tumors from snRNA-seq, RNA counts of each sample were aggregated into pseudobulk data. Raw read counts were then subjected to the DESeq2 package using default settings. Significantly differentially expressed genes were filtered for using a $p$-adj <0.05 cut-off. Wald test was used to get a raw $p$ value and then applied Benjamini–Hochberg (BH) method to produce $p$-adj (Fig. 2e)

## Pseudotime trajectory analysis

To explore potential stages of plasticity between RP and RPC tumors, pseudotime trajectory analysis was performed using Monocle2 (version 2.22.0). All samples were merged into one Seurat object and then converted into CellDataSet object following official vignette of Monocle. After size factor and dispersion estimation, 200 variable genes of tumor cells extracted by SelectIntegrationFeatures of Seurat were applied to setOrderingFilters. Dimension reduction was performed by setting max_component to 2 and method to DDRTree. Cells were then ordered along an unsupervised pseudotime trajectory. Trajectories were plotted using the plot_cell_trajectory function of Monocle. A heatmap visualizing expression of the 200 variable genes along the pseudotime trajectory was made using the plot_pseudotime_heatmap function.

## AUCell binary scores

To map neurogenesis similarities onto RP and RPC snRNA-seq data, the activity of different normal gene expression sets during dentate gyrus

neurogenesis (RGL, nIPC, neuroblast 1)[37] was quantified using AUCell[36]. Binary AUCell score activities were plotted on the UMAP plot with the following cut-off: RGL > 0.065(on), nIPC>0.1(on), neuroblast 1 > 0.075(on).

## Computation of RP and RPC profiles enriched in human SCLC patients

Expression profiling of differentially-expressed genes from our RP versus RPC snRNA-seq pseudo-bulk data were applied to a published human SCLC bulk-RNA seq dataset[38] only using SCLC-A and SCLC-N patients. Mouse genes were first converted into human orthologous by BioMart[75]. Lowly expressed transcripts were excluded from further analysis (FPKM sum <100). Each gene was normalized to this gene's median FPKM across patients. Normalized values were log10 transformed (FPKM+1) and plotted as a heatmap.

## Bioinformatic analysis of human RNA-seq and WGS datasets

Human bulk RNA sequencing samples were assembled from published studies including samples of SCLC[20], LUADs and non-tumor tissue[76]. Raw data were processed as described previously[77]: reads were aligned to the human reference genome hg38 with STAR aligner[78] and gene-level expression was quantified as transcripts per million (TPM) using RSEM[79] before CASP8 expression levels were plotted as log2(TPM+1). Correlations of CASP8 transcript levels and genome alterations was performed for $n = 71$ patients referring to matched paired-end RNA-seq and WGS data, respectively[20]. As previously described[20], expression levels were determined with Cufflinks and quantified as fragments per kilobase of exon per million fragments mapped (FPKM). CASP8 expression refers to the sum of all transcript variants reported and quantified for this gene locus and to the major transcript variant NM_033356. Genomic alterations of CASP8 were studied as mutations and copy number alterations. As previously reported[20], somatic mutations of CASP8 are rare and were found in 2 patients. Chromosomal copy number changes of the locus encompassing CASP8 were not focal and refer to larger chromosomal arm-level losses or gains. Copy number states were determined as integral copy number (iCN) following corrections for tumor purity[20] (Fig. 1f). Bulk RNA sequencing data of 77 human NSCLC cell lines and 14 SCLC cell lines obtained from Expression Atlas were log2 transformed (TPM+1) and plotted for expression of caspase 8. Bulk RNA sequencing data of 5 primary mouse NSCLC cell lines and 10 primary mouse SCLC cell lines both from C57BL/6 background were log2 transformed (TPM+1) and plotted for expression of caspase 8.

## RP- and RPC-derived cell line bulk RNA-Seq data analysis

Primary data analysis was conducted using the rnaseq pipeline from the nf-core suite (v3.7)[80]. Sequencing reads were aligned to the GRCm39 (v105) mouse reference genome using the STAR aligner (v2.7.10a)[78]. Gene quantification was conducted using Salmon (v1.5.2)[81]. The pipeline was executed with default parameters. Downstream differential expression analysis was performed using DESeq2 (v1.36.0)[82], also with default parameters. To correct for the deviation of the original raw $p$-values from the expected uniform null distribution, we used fdrtool (v1.2.17)[82] setting the stat parameter to "normal" (default). Subsequently, the Benjamini–Hochberg procedure was applied to correct the $p$-values for multiple tests. GO enrichment analysis was conducted using gprofiler2 (v0.2.1)[83]. The selection criteria focused on differentially expressed genes with log2FoldChange > 0.58 and adjusted $p$-value < 0.05, using ordered gene query and setting the correction_method parameter to "fdr" to adjust for multiple tests. Heatmaps of top differentially expressed genes were generated using the R package pheatmap and taking the top 50 up- or downregulated genes, respectively. Genes were ranked by their adjusted $p$-value, but only genes with absolute log2FoldChange > 2 were included in the heatmap. Expression values were scaled row-wise

(z-score). For Venn diagrams, GO enrichment of DMR genes was performed using gprofiler2, while selecting only genes with absolute Mean_Diff > 10 (Mean_Diff reflecting the mean difference in methylation levels) for both hypo- and hypermethylated genes. Only terms with size ≥10 were considered. Fisher's exact test was used to evaluate whether the overlap of enriched GO terms between methylation and RNA-Seq assays exceeds that expected by chance. As this is an across-assay comparison, the background universe was defined as the union of all GO terms tested in at least one of the assays, matching the Venn diagram representation shown in Fig. 3d, e.

## Immunohistochemistry

For immunohistochemistry, tissues were fixed in 4% Formaldehyde solution (Merck) and subsequently embedded in paraffin. After sectioning (4 μm) tissue was deparaffinized and rehydrated according to standard procedure. For antigen retrieval samples were heated to 100 °C for 30 min, in a citrate-based buffer (Vector Laboratories, H-3300) for cleaved caspase 3 stainings or in a TE-based buffer [10 mM Trizma Base, 1 mM EDTA, pH = 9] in a pressure cooker for ASCL1 stainings. Subsequently, samples were blocked for endogenous peroxidases (BLOXALL Endogenous Peroxidase and Alkaline Phosphatase Blocking Solution, Vector Laboratories, SP-6000, 15 min) and for unspecific binding with Avidin/Biotin (Avidin/Biotin Blocking Kit, Vector Laboratories, SP-2001, 15 min). For ASCL1 stainings, the mouse-on-mouse blocking buffer (Abcam) was used for 30 min according to the protocol. Samples were incubated overnight at 4 °C with anti-cleaved caspase 3 antibody (Cell Signalling 9661) 1:500 in blocking buffer (PBS, 1% BSA, 0.003% NaN3, 0.05% Tween20) or anti-ASCL1 (BD Pharmingen, 556604) 1:300 in TBS 1% BSA. The following day, samples were washed three times in PBS-T and incubated with secondary antibody anti-rabbit Biotin (Perkin Elmer, NEF813) 1:1000 in blocking buffer for 1 h (for cleaved caspase 3 stainings) or horse radish peroxidase (HRP) polymer detector for 15 min. for anti-ASCL1 (Abcam, mouse on mouse kit) and washed as before. For the cleaved caspase 3 staining, samples were incubated with PBS plus 1/60 Biotin, plus 1/60 Avidin (VECTASTAIN® Elite® ABC HRP Kit, Vector Laboratories, PK-6100) for 30 min. Both stainings were developed using DAB chromogen (Abcam, ab6423) according to the manufacturer's instructions and counterstained using Hematoxylin. Sections were dehydrated using increasing ethanol concentrations and fixed in xylene. The slides were mounted in Entellan. Sections were also stained for Ki-67 (Cell Marque 275-R10), CD45 (BD 550539), CD31 (DIA-310-M, Dianova), CD56 (Zytomed RBK050) and phosho-γH2AX (Cell Signaling 9718). For Oil Red O staining and visualization of lipid droplets, 5 mg/ ml Oil Red O stock solution was prepared by dissolving it in isopropanol. Oil Red O stock solution was added to distilled water in a ratio of 3:2, kept for 10 min and filtered using Whatman filter paper and 0.45-μm syringe filter. Next, Oil Red O working solution was added to the fixed cells for 10 min, rinsed with distilled water and washed three times.

## TUNEL staining

The DeadEnd™ Fluorometric TUNEL System kit (Promega, #G3250) was used for TUNEL staining and was performed according to manufactures protocol for paraffin-embedded tissue. Propidium iodide step was omitted and slides were instead covered with ProLong™ Gold Antifade Mounting medium with DAPI (Invitrogen™, #P36941). Fluorescence pictures were obtained performed with Keyence BZ-X800 Microscope and analysis were performed with BZ-X800 Analyzer software.

## Immunoblotting

Lysates from mouse and human SCLC cell lines were prepared using IP-lysis buffer (30 mM Tris-HCl, 120 mM NaCl, 2 mM EDTA, 2 mM KCl, 1% Triton-X-100, pH 7.4, Protease and Phosphatase inhibitor). Samples were centrifuged at 11,000 × $g$ for 20 min at 4 °C and then further used

for Western Blotting. Lysate concentrations were adjusted to equal protein concentrations using the bicinchoninic acid (BCA) protein assay (Biorad). Samples were heated to 80 °C for 10 min, were separated via gel electrophoresis and transferred to nitrocellulose membranes (Bio-Rad). The membranes were blocked in PBS with 0.1% Tween 20 (PBST) with 5% (w/v) dried milk powder for at least 1 h and incubated with primary antibodies (caspase 8, Enzo Life Sciences ALX-804-447-C100 1:1000, ß-Actin, Sigma A1978, 1:10,000, a-Tubulin, Santa Cruz sc-5286 1:10,000, HSP90, Cell Signaling 4874, Vinculin, Cell Signaling 13901, 1:1000, ASCL1, BD Pharmingen 556604, 1:1000, YAP1, Cell Signaling 4912, 1:1000, REST1, Thermofischer bs2590, 1:1000, phospho-MLKL(S345) 37333S, 1:1000, MLKL, Millipore MABC604 1:1000, RIPK1, Cell Signaling 3493, 1:1000, RIPK3, Cell Signaling 15828, 1:1000, phospho-γH2AX, Cell Signaling 9718, 1:1000, overnight at 4 °C. After washing with PBST, membranes were incubated with 1:10,000 diluted HRP-coupled secondary antibodies for at least 1 h at room temperature. After another washing step, bound antibodies were detected using chemiluminescent substrate Immobilon Luminata Classico (Millipore) or chemiluminescent Amersham ECL Prime Western Blotting Detection Reagent (RPN2235, Cytiva) and developed with X-ray films CL-XPosure™ (Thermo Scientific) or the FUSION Solo S system and software (Vilber). Membranes were stripped if necessary, using stripping buffer (ThermoScientific, 21059) for 15 min at RT. All western blot images are representative of at least two independent experiments.

### ELISA

For determining TNF, IL6, IL10 and TGF-beta protein concentration in whole lung tissue extracts, 300 μl of Tissue Lysis Buffer (10 mM Tris-HCl, 150 mM NaCl, 1%NP-40, 10% Glycerol, 5 mM EDTA) was added to 5 mg of snap-frozen tissue. Mixture was immediately homogenized with ceramic bids in a peqlab vial for 2 ×30 s using the Precellys 24-dual homogenizer (Peqlab) and the supernatant was collected after centrifuging at 13,000 for 20 min. Lysate was aliquoted and was kept at −20 °C until analyzed. ELISA assays (Mouse TNF-alpha DuoSet ELISA, D410, RnD systems, Mouse IL-6 DuoSet ELISA, D406, RnD systems, Mouse IL-10 DuoSet ELISA, D417, RnD systems, Mouse TGF-beta 1 DuoSet ELISA, DY1679, RnD systems) were performed according to the manufacturer´s instructions, using equal volumes of lysates, in triplicates. The results were normalized to mg of protein of the lysate (based on BCA measurement).

### Live cell imaging (IncuCyte)

Cells were plated in 96-well plates ($10^4$ cells per well) and stimulated next day with TNF (20 ng/ml), Birinapant (1 μM) (S7015, sellekchem), Cisplatin (20 μM), Etoposide (10 μM). Cells were imaged using the 10× objective within the IncuCyte SX5 live cell imaging system (Sartorius). For dead cell quantification, 100 nM DRAQ7 (Thermofisher) was added to each well. For cleaved caspase 3 (CC3), cells were co-incubated with 5 μM CellEvent™ Caspase3/7 Detection Reagent (Invitrogen). Cells were imaged for indicated timepoints every 2 h. Analysis for confluence, DRAQ7- positive (dead) or CC3 (apoptotic) was performed using the Software IncuCyte 2021A and 2022B (Sartorius).

### FACS

In order to isolate immune cells from lungs or lung tumors, whole tissue was dissected and minced with scalpels into fragments small enough to be aspirated into a 5 ml pipette at RT. 45 ml of tissue suspension was incubated with 5 ml of a 10× Triple Enzyme Mix (1 g Collagenase IV, 100 mg Hyaluronidase and 20,000 Units DNase IV into 80 ml HBSS) at RT for 90 min on a shaker at 80 rpm. Cell suspension was repeatedly pipetted to further dissociate cells, centrifuged at 50 × g at RT for 10 min and the supernatant was collected

by passing it through a 70 μm nylon strainer. The bigger pellets in the bottom of the tube were then discarded, and the filtered supernatant was centrifuged at 200 × g for 5 min. Cell pellets were washed with 10 ml Wash Buffer (1 g BSA and 2 ml 0.5 M EDTA in 800 ml HBSS) at 200 × g for 5 min once and were resuspended with 2 ml ACK lysing buffer (Gibco) for 1 min to deplete red blood cells. Cells were washed with PBS and immediately stained for live/dead cells using the Fixable Viability Dye eFluor 660 or eFluor 450 (eBioscience) (1:1000) in PBS for 30 min, at 4 °C. Cells were then washed twice with FACS buffer (PBS, 2% FCS) and stained with the respective FACS antibodies (1:1000) for another 30 min, at 4 °C. Fc block (CD16/32, biolegend, 1:50) was used 15 min before adding the antibodies for blocking Fc receptors. For subsequent intracellular stainings, cell pellets were resuspended in 200 μl Fixation/Permeabilization buffer (eBioscience) and incubated overnight at 4 °C. The next day, cells were washed with 1× Permeabilization buffer (eBioscience) and incubated for 15 min with 2% goat serum before adding the respective intracellular antibodies 1:50, for 30 min at 4 °C in 1× Permeabilization buffer. After washing twice with 1× Permeabilization buffer cells were resuspended in FACS buffer. Measurements were acquired using a BD LSR Fortessa flow cytometer, and data were analyzed with the FlowJo (10.6.1) software. Respective FACS antibodies: CD45-PE (30.F11) and CD45-FITC (30.F11) biolegend, CD4-V450 (RM4-5) BD Horizon, CD25-PE-Cy7 (PC61.5) ebioscience, CD11b-PE (M1/70) ebioscience, CD44-PE-Cy7 (IM7) ebioscience, CD62L-PE (MEL-14) ebioscience, CD3-FITC (145-2C11) ebioscience, CD8-PE (53-6.7) biolegend, CD69-PE (H1.2F3), ebioscience, CD19-BV711 (1D3), BD Horizon, Gr1 (Ly6G/Ly6C) (RB6-BC5), ebioscience, PD1-FITC (J43) ebioscience, TIM3-APC (RMT3-23) ebioscience, NK1.1-FITC (PK-136) biolegend, RORγc-PE (AFKJS-9) ebioscience, CD206-BV421 (C068C2) and Rat IgG2a, κ-isotype Ctrl -BV421 (RTK-2758) biolegend, foxp3-PE (FJK-16s) and Rat IgG2a kappa Isotype Control, PE (eBR2a) ebioscience.

### Statistics and reproducibility

All experiments throughout were performed as biological replicates (n) at least 2–3 times in each cell line. Graphical data are shown as means calculated between these biological replicates ± standard error mean. Representative image data are shown from at least two independent biological replicates. All in vivo mouse experiments were repeated in two or three independent cohorts with at least three independent biological samples in each cohort. The results from all different cohorts were pooled to be shown in figures. No data were excluded from the analyzes. The data were analyzed by GraphPad Software and either t-tests or one-way ANOVA was used to calculate p values as indicated. p values were considered significant at p < 0.05. Mantel–Cox test was used for survival curves and tumor incidence over time. Fisher´s exact test on actual number of mice above or below indicated cut-offs was used for contingency tables (Figs. 1h–j; 4g, i; 5d, e).

### Reporting summary

Further information on research design is available in the Nature Portfolio Reporting Summary linked to this article.

## Data availability

Patient derived MeDIP-seq data has been deposited at the European Genome-phenome Archive (EGA), which is hosted by the EBI and the CRG, under accession number EGAS50000000506. Further information about EGA can be found at https://ega-archive.org and "The European Genome-phenome Archive of human data consented for biomedical research". Methyl-seq data from RP- and RPC-derived cell lines are available from Gene Expression Omnibus (GEO) under accession number GSE274232. Bulk-RNA-seq data from RP- and RPC-

derived cell lines are available from GEO, accession number GSE271260 and snRNA-seq data from frozen RP- and RPC- mouse tumors are available from GEO, accession number GSE274809. Source data are provided with this paper.

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

## Acknowledgements

This work was supported in part from the German Research Foundation grant SPP2306, project ID 461704389 (SvK); the German Research Foundation grant CRC1399 ID 413326622 (SvK, FB, HCR, MP, JG, RKT, JB, RH-H, HG, AQ, MRS, MPe); the German Research Foundation grant CRC1403 ID 414786233 (SvK, MP); the German Research Foundation grant CRC1310 ID 325931972 (SvK); the German Research Foundation grant CRC1430 ID 424228829 (HCR); the German Research Foundation grant CRC1530 ID 455784452 (SvK, HCR); the German Research Foundation grant HA 8562/4-1 (RH-H); the German Research Foundation grant RU5504 HA 8562/5-1 (RH-H); the Federal Ministry of Education and Research (BMBF) InCa-01ZX2201A (SvK, RT, JG, HCR, MP, MPe); the Ministry of Culture and Science of the State of Northrhine Westphalia, CANcer TARgeting (CANTAR) (SvK, JG, RKT, RH-H, HCR, MP); the Center for Molecular Medicine Cologne (CMMC) A07 (SvK); the German Cancer Aid (Mildred Scheel Nachwuchszentrum) (FB, JB); the Center for Molecular Medicine Cologne (CMMC) Junior Research Group (RH-H); the Fritz Thyssen Foundation 10.22.1.010MN (RH-H). RKT was funded through TACTIC (70115201) by the German Cancer Aid. We thank Marcel Schmiel and Laura Kaiser for helpful information on single cell analysis, Lukas Sebeke for help with the MRI imaging, Elisa Motori for advice and Alexandra Florin for technical support. We furthermore thank the Regional Computing Center of the University of

Cologne (RRZK) for providing computing time on the DFG-funded (Funding number: INST 216/512/1FUGG) High Performance Computing (HPC) system CHEOPS, as well as support. Figure 5g was generated using Biorender.com.

## Author contributions

A.A. and S.v.K. designed experiments and wrote the manuscript with input from all co-authors, A.A. carried out most of the experiments, analyzed data and supervised L.M., Ju.B. J.S. and F.D. who assisted with experiments. F.L. performed experiments and processed and analyzed sc-RNA sequencing data, C.M.B. performed experiments, analyzed data, and assisted with mouse work, I.K., V.S., P.H., P.N. and R.H.H. set up snRNA-seq library preparation and pre-analysis of snRNA-seq raw data. A.D. and F.I.Y. performed experiments and assisted with mouse work, L.K. crossed and analyzed RP-Mlkl$^{AA/AA}$ mice, C.G., A.P. M.R.S. performed and analyzed methyl-seq experiments, A.S. and H.C.R. performed initiated initial mouse crosses, S.T. performed and analyzed spheroid assays and differentiation experiments L.A.T.F. analyzed human copy number changes in casp8, A.T.A. analyzed RP-derived cell line RNA-seq data, J.B. analyzed human TCGA datasets, H.G. and T.P. assisted with MRI imaging, A.Q. performed pathological inspections, M.Pe analyzed human datasets, R.K.T. and J.G. provided methyl-Seq data from human SCLC patients, M.P. provided mice and critical input for experimental design and F.B. established an snRNA-seq pipeline, ran the pre-analysis, provided bioinformatic analysis for human SCLC cases and critical input.

## Funding

## Competing interests

H.C.R. received consulting and lecture fees from AbbVie, AstraZeneca, Roche, Janssen-Cilag, Novartis, Vertex and Merck. H.C.R. received research funding from Gilead and AstraZeneca. H.C.R. is a co-founder of CDL Therapeutics GmbH. R.K.T. was a founder and shareholder of and consultant to PearlRiver Bio (now part of Centessa), a shareholder of Centessa and founder, shareholder and CEO of DISCO Pharmaceuticals. All other authors declare that they have no competing interests.

## Additional information

¹Department of Translational Genomics, University of Cologne, Faculty of Medicine and University Hospital Cologne, Cologne, Germany. ²CECAD Cluster of Excellence, Faculty of Medicine and University Hospital Cologne, Cologne, Germany. ³Center for Molecular Medicine Cologne, Faculty of Medicine and University Hospital Cologne, Cologne, Germany. ⁴Institute for Genetics, University of Cologne, Cologne, Germany. ⁵Institute for Translational Epigenetics, University of Cologne, Faculty of Medicine and University Hospital Cologne, Cologne, Germany. ⁶Institute of Medical Statistics and Computational Biology, Faculty of Medicine, University of Cologne, Cologne, Germany. ⁷CECAD Cluster of Excellence, Faculty of Mathematics and Natural Sciences, University of Cologne, Cologne, Germany. ⁸Department I of Internal Medicine, Faculty of Medicine and University Hospital Cologne, Cologne, Germany. ⁹Department of Radiology, University of Cologne, Faculty of Medicine and University Hospital Cologne, Cologne, Germany. ¹⁰Institute of Pathology, University of Cologne, Faculty of Medicine and University Hospital Cologne, Cologne, Germany. ¹¹Faculty of Medicine and University Hospital Cologne, University of Cologne, Mildred Scheel School of Oncology (MSSO), Cologne, Germany. ¹²Department of Hematology and Stem Cell Transplantation, University Hospital Essen, University Duisburg-Essen, German Cancer Consortium (DKTK partner site Essen), Essen, Germany. ¹³Department of Otorhinolaryngology, Head and Neck Surgery, University Hospital of Cologne, Faculty of Medicine and University Hospital Cologne, Cologne, Germany. ✉e-mail: s.vonkarstedt@uni-koeln.de

