## [Transparent Peer Review file · Nature Communications]

Lack of Caspase 8 Directs Neuronal Progenitor-like reprogramming and Small Cell Lung Cancer Progression

Corresponding Author: Professor Silvia Von Karstedt

Version 0:

Reviewer comments:

Reviewer #2

(Remarks to the Author)

The authors have made an impressive effort to address all of my previous comments, resulting in a significantly improved manuscript. I am pleased to see the inclusion of new data and additional controls, which provide important mechanistic insights. Overall, this is an excellent manuscript that introduces an exciting new model for studying SCLC. I strongly support its publication.

Reviewer #4

(Remarks to the Author)

In this manuscript, Androurlidaki et al report their discoveries regarding the role of loss of Caspase8 in the pathobiology of small cell lung cancer.

As a primary model system, the authors generated a new mouse model of SCLC by incorporating genetic loss of Caspase8 on the background of the classic SCLC model driven by p53 and Rb inactivation. Using this model, they performed extensive pathologic, transcriptomic, epigenomic and immunologic studies to provide a compelling view of the complex role of caspase8 in mediating necroptosis, pre-cancerous inflammation and immune dysregulation culminating in differentiation defect, and an aggressive & metastatic SCLC phenotype.

The study is well executed and controlled. The data is of high quality and the conclusions are largely supported by the data provided. The findings are of high interest to the broad cancer research community, especially in the light of increasing recognition of neuroendocrine differentiation as a mechanism of resistance development from multiple targeted therapies. The manuscript would benefit from extensive editing to improve its readability and transmission of key messages. This is particularly important in the description of gene expression and epigenomic results, GO term analysis and pathway enrichment analysis which at times are difficult to follow and overly verbose. Finally, the discussion should also be significantly edited to enhance readability.

Some concerns and suggestions:

- 1) A key weakness of the study is that the model of homozygous Caspase 8 inactivation does a poor job of mimicking the reduced expression of Caspase 8 seen in humans. The authors need to highlight this weakness as a major caveat in the discussion of their findings to set an appropriate "tone" for translation of findings to the human condition.
- 2) Authors describe a primarily epigenetic basis for reduced expression of Caspase 8 in SCLC. However, based on Fig 1a and extended data Fig 1a, the reduction in protein level is more pronounced than transcript level. How do they reconcile this? It is likely that casp8 dysregulation at the translational or posttranslational levels also occur in SCLC. The authors are encouraged to discuss this possibility.
- 3) Kaplan-Meier curves did not show significant difference between RP and RPC cohorts despite increase in number of tumors and metastasis in the RPC group. What could the underlying reason be? Again, a discussion of this point is warranted. This can be supported by, when possible, a discussion of the immediate cause of mortality of these mice.
- 4) Authors need to perform formal statistical tests of significance for Fig 3d-e to show relationship between hypermethylated promoters and downregulation of gene expression. These tests are required to show the 72 genes showing overlapping change in methylation and RNA expression is more than expected by chance occurrence.
- 5) The expression data in figure 5f-g for RP, RPC, RP-Mik1AA/AA, RPC-MikAA/AA need to be combined into one plot for

each gene so that direct comparison can be made of the impact of Mik1AA/AA allele.

Reviewer #5

(Remarks to the Author)

Overall, this is a well-integrated study linking low caspase-8 expression to neuroprogenitor-like reprogramming and metastatic progression in small cell lung cancer. The authors have addressed the reviewer's previous comment comprehensively.

In late-stage tumors (Figs. 1–3), endpoint analyses in the mouse model are used to demonstrate neuronal and stem-like features of RPC mice, whereas early-stage lesions (Figs. 4–6) are employed to probe mechanistic insights, particularly those involving Tregs. This creates some conceptual disconnect between the two stages, although the authors acknowledge and discuss this in the revised version.

One additional suggestion: the section on DNA methylation remains underdeveloped, with only a single heatmap shown in Figure 3 and limited functional characterization. I recommend de-emphasizing this component throughout the text and potentially removing it from the summary schematic to maintain focus on the core mechanistic findings, as no further functional data are shown.

Reviewer #6

(Remarks to the Author)

Point by point reply to reviewer comments

Reviewer #2 (Remarks to the Author):

The authors have made an impressive effort to address all of my previous comments, resulting in a significantly improved manuscript. I am pleased to see the inclusion of new data and additional controls, which provide important mechanistic insights. Overall, this is an excellent manuscript that introduces an exciting new model for studying SCLC. I strongly support its publication.

We sincerely thank reviewer for helping us improve the manuscript.

Reviewer #4 (Remarks to the Author):

In this manuscript, Androulidaki et al report their discoveries regarding the role of loss of Caspase8 in the pathobiology of small cell lung cancer.

As a primary model system, the authors generated a new mouse model of SCLC by incorporating genetic loss of Caspase8 on the background of the classic SCLC model driven by p53 and Rb inactivation. Using this model, they performed extensive pathologic, transcriptomic, epigenomic and immunologic studies to provide a compelling view of the complex role of caspase8 in mediating necroptosis, pre-cancerous inflammation and immune dysregulation culminating in differentiation defect, and an aggressive & metastatic SCLC phenotype.

The study is well executed and controlled. The data is of high quality and the conclusions are largely supported by the data provided. The findings are of high interest to the broad cancer research community, especially in the light of increasing recognition of neuroendocrine differentiation as a mechanism of resistance development from multiple targeted therapies.

The manuscript would benefit from extensive editing to improve its readability and transmission of key messages. This is particularly important in the description of gene expression and epigenomic results, GO term analysis and pathway enrichment analysis which at times are difficult to follow and overly verbose. Finally, the discussion should also be significantly edited to enhance readability.

We thank the reviewer for appreciating our study. We understand that some parts were overly descriptive and hard to follow. We have now edited significant parts of the results text as well as the discussion to enhance readability.

Some concerns and suggestions:

1) A key weakness of the study is that the model of homozygous Caspase 8 inactivation does a poor job of mimicking the reduced expression of Caspase 8 seen in humans. The authors need to highlight this weakness as a major caveat in the discussion of their findings to set an appropriate tone for translation of findings to the human condition.

While we fully agree with the reviewer that human SCLC does not present with genetic caspase 8 deficiency), there is ample data in the field showing that SCLC in the majority of cases lacks caspase 8 protein expression. This is supported by a body of literature on epigenetic silencing of the caspase 8 promotor region in human SCLC cell lines not allowing for its transcription¹⁻³. A few years earlier a striking difference in pro-caspase-8 protein expression comparing

SCLC with NSCLC lines was noted in 80% of cell lines tested⁴. As a functional consequence, human SCLC cell lines were shown to be entirely resistant to extrinsic apoptosis^{5,6}. Importantly, functional heterozygosity of the caspase 8 locus *in vivo* still allows for extrinsic apoptosis to proceed⁷ indicating that heterozygous caspase 8 deletion in RP mice would not functionally mimic the situation found in human SCLC cell lines. Together, these data suggest that homozygous deletion of caspase 8 represents the best approximation of the lack of caspase 8 protein expression in human SCLC.

In a proteogenomic dataset on human SCLC tissue with adjacent normal lung⁸ we could confirm that caspase 8 protein levels are low-to-absent in SCLC as compared to normal lung (Fig. 1a). We also analyzed tumor purity in human SCLC bulk RNA sequencing data⁹ and find that caspase 8 mRNA expression correlates with tumor purity indicating a high likelihood for the residual mRNA signal to stem from normal cell contaminants within human biopsies (Extended Data Fig. 1b). Lastly, we could also not detect any caspase 8 protein expression in human SCLC cell lines used in the study (Extended Data Fig. 1d).

While we cannot rule out that in certain cases caspase 8 expression is reactivated because of epigenetic reprogramming in human SCLC cell lines exposed to prolonged *in vitro* culture, the collective data available suggest functional absence of caspase 8 in the majority of human SCLC across our own data and prior publications on the topic. Yet, to make this point clearer and highlight limitations of our study (homozygous deletion to mimic lack of protein expression caused by epigenetic regulation) we have dedicated a new section to this in the discussion.

2) Authors describe a primarily epigenetic basis for reduced expression of Caspase 8 in SCLC. However, based on Fig 1a and extended data Fig 1a, the reduction in protein level is more pronounced than transcript level. How do they reconcile this? It is likely that casp8 dysregulation at the translational or posttranslational levels also occur in SCLC. The authors are encouraged to discuss this possibility.

We thank the reviewer for pointing this interesting observation out. Indeed, the more prominent reduction of protein levels of caspase 8 might suggest translational or post-translational regulation in addition to epigenetic silencing. It is interesting to note that caspase 8 stability was shown to be promoted by cullin 3-mediated K63 polyubiquitination¹⁰ and neuroendocrine cancers seem to overexpress cullin 3¹¹.

Yet, given the fact that data in Fig 1a and extended data Fig 1a stem from human biopsy samples, which contain tumor cells and other non-transformed microenvironmental cells (immune cells, normal alveolar cells etc.), it is difficult to deduct in this dataset (for which we do not have tumor purity information) from which cell type in the mixed sample this discrepancy comes from. We have added a section to the discussion regarding this point.

3) Kaplan-Meier curves did not show significant difference between RP and RPC cohorts despite increase in number of tumors and metastasis in the RPC group. What could the underlying reason be? Again, a discussion of this point is warranted. This can be supported by, when possible, a discussion of the immediate cause of mortality of these mice.

Indeed, we also expected increased metastasis to result in decreased survival. Yet, this was not the case. Notably, while RPC mice presented with multiple tumors and very frequent metastasis, these tumors were clearly smaller than in RP mice at experimental endpoint. This suggests that the cause of death of RPC mice was likely metastasis and in RP mice limited lung functionality due to big primary lung tumors. Hence, Kaplan Meyer survival in our study likely reflects two distinct causes of death in the two groups. The fact, however, that at 40

weeks post inhalation, significantly more RPC mice than RP had died, suggests that, at that time, differences in metastasis likely determined survival while at experimental endpoint this effect was washed out by primary lung tumor-determined death in the RP group. We have added a short section on this point to the discussion.

4) Authors need to perform formal statistical tests of significance for Fig 3d-e to show relationship between hypermethylated promoters and downregulation of gene expression. These tests are required to show the 72 genes showing overlapping change in methylation and RNA expression is more than expected by chance occurrence.

*We thank reviewer for his comment. We would like to clarify that in the Venn diagrams shown, number of overlapping pathways are plotted and not number of genes. We have made every effort to now clearly state this in the text and figure legends. On these overlapping pathways, we have now performed statistical tests and indeed we could show strong significance in the overlap between hypermethylated pathways and downregulated ones (****p=2.54e-23), as well as in the overlap between hypomethylated and upregulated ones (****p=2.68e-08).*

5) The expression data in figure 5f-g for RP, RPC, RP-Mik1AA/AA, RPC-Mik1AA/AA need to be combined into one plot for each gene so that direct comparison can be made of the impact of Mik1AA/AA allele.

We agree and have now combined all genotypes into one plot for each gene (new Fig. 5f).

Reviewer #5 (Remarks to the Author):

Overall, this is a well-integrated study linking low caspase-8 expression to neuroprogenitor-like reprogramming and metastatic progression in small cell lung cancer. The authors have addressed the reviewer's previous comment comprehensively.

In late-stage tumors (Figs. 1-3), endpoint analyses in the mouse model are used to demonstrate neuronal and stem-like features of RPC mice, whereas early-stage lesions (Figs. 4-6) are employed to probe mechanistic insights, particularly those involving Tregs. This creates some conceptual disconnect between the two stages, although the authors acknowledge and discuss this in the revised version.

One additional suggestion: the section on DNA methylation remains underdeveloped, with only a single heatmap shown in Figure 3 and limited functional characterization. I recommend de-emphasizing this component throughout the text and potentially removing it from the summary schematic to maintain focus on the core mechanistic findings, as no further functional data are shown.

We thank the reviewer for pointing this out. Indeed, we have not gone into providing further mechanistic insights regarding this part. We believe that regulation of methylation is an important feature of the RPC phenotype but since further studies are indeed needed to functionally characterize these findings, we have now de-emphasized the methylation component throughout the manuscript and removed it from the scheme as suggested.

Reviewer #6 (Remarks to the Author):

We thank the reviewer for their contribution which helped improve the manuscript.

Literature:

1. Shivapurkar, N. *et al.* Differential inactivation of caspase-8 in lung cancers. *Cancer Biol. Ther.* **1**, 65–69 (2002).
2. Shivapurkar, N. *et al.* Loss of expression of death-inducing signaling complex (DISC) components in lung cancer cell lines and the influence of MYC amplification. *Oncogene* **21**, 8510–8514 (2002).
3. Belyanskaya, L. L. *et al.* TRAIL-induced survival and proliferation of SCLC cells is mediated by ERK and dependent on TRAIL-R2/DR5 expression in the absence of caspase-8. *Lung Cancer Amst. Neth.* **60**, 355–365 (2008).
4. Joseph, B., Ekedahl, J., Sirzen, F., Lewensohn, R. & Zhivotovsky, B. Differences in expression of pro-caspases in small cell and non-small cell lung carcinoma. *Biochem. Biophys. Res. Commun.* **262**, 381–387 (1999).
5. Bebbber, C. M. *et al.* Ferroptosis response segregates small cell lung cancer (SCLC) neuroendocrine subtypes. *Nat. Commun.* **12**, 2048–19 (2021).
6. Hopkins-Donaldson, S. *et al.* Silencing of death receptor and caspase-8 expression in small cell lung carcinoma cell lines and tumors by DNA methylation. *Cell Death Differ.* **10**, 356–364 (2003).
7. Kang, T.-B. *et al.* Caspase-8 serves both apoptotic and nonapoptotic roles. *J. Immunol. Baltim. Md 1950* **173**, 2976–2984 (2004).
8. Q, L. *et al.* Proteogenomic characterization of small cell lung cancer identifies biological insights and subtype-specific therapeutic strategies. *Cell* **187**, (2024).
9. Cun, Y., Yang, T.-P., Achter, V., Lang, U. & Peifer, M. Copy-number analysis and inference of subclonal populations in cancer genomes using Sclust. *Nat. Protoc.* **13**, 1488–1501 (2018).
10. Jin, Z. *et al.* Cullin3-Based Polyubiquitination and p62-Dependent Aggregation of Caspase-8 Mediate Extrinsic Apoptosis Signaling. *Cell* **137**, 721–735 (2009).
11. Park, J.-U. *et al.* The differentially expressed gene signatures of the Cullin 3-RING ubiquitin ligases in neuroendocrine cancer. *Biochem. Biophys. Res. Commun.* **636**, 71–78 (2022).